# Harmonisation and trends of 20-years tropical tropospheric ozone data

Elpida Leventidou[1], Mark Weber[1], Kai-Uwe Eichmann[1], and John P. Burrows[1]

[1]Institute of Environmental Physics (IUP), University of Bremen, Germany

*Correspondence to:* E. Leventidou (Levent@iup.physik.uni-bremen.de)

**Abstract.** Using the convective clouds differential (CCD) method on total ozone and cloud data from three European satellite instruments GOME/ERS-2 (1995–2003), SCIAMACHY/Envisat (2002–2012), and GOME-2/MetOp-A (2007–2015) it is possible to retrieve tropical tropospheric columns of ozone (TTCO) which are in good agreement with in-situ measurements. Small differences in TTCO between the individual instruments are evident and therefore the individual datasets retrieved are harmonised into one consistent time-series starting from 1996 until 2015. Correction offsets (bias) between the instruments using SCIAMACHY as intermediate reference have been calculated and six different harmonisation scenarios have been tested. Finally, the datasets have been harmonised applying no correction to GOME data while GOME-2 has been corrected using for each grid-box the mean bias with respect SCIAMACHY for the years of common operation (2007–2012). Depending on the choice of harmonisation, the magnitude, pattern, and uncertainty of the trend can strongly vary. The harmonisation represents an additional source of uncertainty in the merged dataset and derived trend estimates. For the preferred harmonised dataset, the trend ranges between -4 and 4 DU decade$^{-1}$. The trend of the tropically averaged tropospheric ozone is equal to 0±0.64 DU decade$^{-1}$ ($2\sigma$). Regionally, tropospheric ozone has a statistically significant increase by ∼3 DU decade$^{-1}$ over southern Africa (∼1.5 % year$^{-1}$), the southern tropical Atlantic (∼1.5 % year$^{-1}$), southeastern tropical Pacific Ocean (∼1 % year$^{-1}$), and central Oceania (∼2 % year$^{-1}$). Additionally, over central Africa (2–2.5 % year$^{-1}$) and south India (∼1.5 % year$^{-1}$), tropospheric ozone increases by ∼2 DU decade$^{-1}$. These regional positive tropospheric ozone trends maybe linked to anthropogenic activities such as emissions in mega cities or biomass burning in combination with changes in meteorology or/and long range transport of precursor emissions. On the other hand, tropospheric $O_3$ decreases by ∼-3 DU decade$^{-1}$ over the Caribbean sea and parts of the North Pacific Ocean (∼ -2 % year$^{-1}$), and by less than -2 DU decade$^{-1}$ over some regions of the southern Pacific and Indian Ocean (∼ -0.5 – -1 % year$^{-1}$). Possible reasons for this decrease are changes in dynamical processes, convection, STE, and precipitation. The comparison of the calculated trends from the current study with tropospheric ozone trends from Heue et al. (2016) and Ebojie et al. (2016) in ten selected mega-cities showed that they agree within $2\sigma$ of the trend uncertainty.

## 1 Introduction

The massive global population growth and urbanization of our societies inevitably leads to increased energy consumption activities related to industry, transportation, and food production. These human activities release a large number of atmospheric

pollutants which can be harmful to public health and/or vegetation and modify the terrestrial climate (Crutzen, 2002). Climate change may also impact air pollution events, since they are both interdependent (WMO/IGAC, 2012). Tropospheric ozone ($O_3$) is regarded as one of the most important surface pollutants due to the fact that it oxidizes the biological tissues causing respiratory problems or even death (WHO, 2006), acts as a greenhouse gas (IPCC, 2007), and controls the oxidizing capacity of the troposphere (Jacob, 2000). $O_3$ in the troposphere is expected to increase by 60 to 80% by 2050 in Southeast Asia, India and Central America under the A2 IPCC (2013) scenario. However, the effects of climate change, especially the increased tropospheric temperatures and water vapour, may offset this increase by 10% to 17% (Stevenson et el., 2000; Grewe et al., 2001; Hauglustaine et al., 2005; IPCC, 2013).

Ozone is not directly emitted in the troposphere but it is a byproduct of the oxidation of volatile organic compounds (VOCs) in the presence of nitrogen oxides ($NO_x$) and sunlight (Jacob, 2000). Young et al. (2013) estimated that 4877 $\pm$ 1706 ($2\sigma$) Tg of $O_3$ are chemically produced every year. Additionally, 477 ± 392 Tg $\cdot yr^{-1}$ are transported from the stratosphere to the troposphere via the stratosphere to troposphere exchange (STE) (Holton and Lelieveld, 1996; Young et al., 2013). Tropospheric ozone loss is controlled by deposition to the Earth's surface and chemical destruction, mainly by photolysis to atomic oxygen ($O(^1D)$), followed by the reaction of $O(^1D)$ with water ($H_2O$) to produce two hydroxyl radicals (2OH) (Jacob, 2000). The net chemical production (production minus loss) is estimated at 618 $\pm$ 550 $Tg\cdot yr^{-1}$ ($2\sigma$) (Young et al., 2013; IPCC, 2013). The mean tropospheric ozone burden is 337$\pm$46 Tg ($2\sigma$) today, which is about 30% more than in 1850 (Young et al., 2013).

The sources of ozone precursors (VOCs and $NO_x$) can be both of anthropogenic and natural origin. Anthropogenic sources of $NO_x$ and VOCs emissions could be fossil fuel combustion, transport, electricity production and industrial processes, agriculture, solvent use and chemical manufacturing. The dominant natural sources of $NO_x$ are lightning, biomass burning, and soil and of VOCs released by several kinds of terrestrial vegetation, mainly forests and shrubs (Jacob, 2000). The spatio-temporal formation of low-level ozone is non-linear and depends on the ratio between $NO_x$ and VOC concentrations which determines the regime of ozone production sensitivity. In a $NO_x$ limited regime (low $NO_x$ and high VOCs), ozone production is insensitive to hydrocarbons and increases when $NO_x$ concentration increases. In a VOCs limited regime (low VOCs and high $NO_x$) a decrease in $NO_x$ results in an increase of ozone concentration whereas a decrease in VOCs decreases ozone (Jacob, 2000; Seinfeld and Pandis, 2006). VOCs limited regime is more likely to exist over urban or industrial regions, while $NO_x$ limited regime is more likely to occur in rural areas downwind of pollution sources (Duncan et al., 2010). For this reason, a successful strategy to mitigate tropospheric ozone pollution effectively requires the knowledge of chemical ozone production regime in order to determine which precursor emissions ($NO_x$ or VOCs) should be controlled.

Various efforts towards reducing $NO_x$ and VOC emissions have been taken in developed countries, particularly in Europe and North America, leading to negative surface ozone trends on a local scale (Derwent et al., 2003; Cooper et al., 2014; Parrish et al., 2014). Nevertheless, tropospheric ozone pollution is a matter of global concern, since ozone and its precursors can be transported from polluted areas to clean regions over continental distances and into the free troposphere through atmospheric dynamics, increasing the tropospheric ozone abundances over remote areas. For example, air masses originated from eastern China have increased ozone abundance over Japan and North America's West Coast, despite the US legislation of reducing $NO_x$ emissions (Parrish et al., 2009; Cooper et al., 2010; Oltmans et al., 2013; Verstraeten et al., 2016). Additionally, the high

tropospheric ozone amounts noticed over the south Atlantic ocean, the so-called "tropical Atlantic paradox", arise from ozone precursor emissions by biomass burning taking place in south America and Africa (Diab et al., 2003; Martin et al., 2002).

The long-term evolution of tropospheric ozone is complex and depends upon the evolution of precursor emissions and climate change. Since the predicted increase of trace gases emissions for the next years is mainly located over low latitudes (Grenfell et al., 2003), long term observations of tropospheric ozone in the tropics should receive particular attention. Various studies have been performed in urban and rural sites using in situ data in order to estimate tropical tropospheric ozone trends. Lelieveld et al. (2004) noticed an increase in surface ozone in the order of 0.4 ppbv year$^{-1}$ over the northeastern tropical Atlantic, 0.4 ppbv year$^{-1}$ over the southeastern tropical Atlantic, and a smaller trend of 0.1 ppbv decade$^{-1}$ over the southwestern tropical Atlantic Ocean, based on ship-borne measurements (1977–2002). Oltmans et al. (2013) using surface and ozonesonde observations noticed an increase of 3.8 % decade$^{-1}$ (0.16 ppbv year$^{-1}$) in Mauna Loa, Hawaii (19.5°N) in the North Pacific since 1974 and a smaller insignificant trend in the order of 0.7 % decade$^{-1}$ (0.01 ppbv year$^{-1}$) in American Samoa (14.5°S) after 1976. Additionally, Cooper et al. (2014) report a significant increase of 0.19 ppbv year$^{-1}$ in the subtropical site of Cape Point in South Africa from 1983 to 2011. Thompson et al. (2014) using ozonesonde data from the SHADOZ stations in Irene (25.9°S, 28.2°W) and Réunion (21.1°S, 55.5°W) noticed statistically significant trends in the middle and upper troposphere of $\sim$ 25 % decade$^{-1}$ (1 ppbv year$^{-1}$) and $\sim$35–45 % decade$^{-1}$ (2 ppbv year$^{-1}$) respectively during winter (June-August). Smaller positive trends appear, close to the tropopause in summer.

Satellite remote sensing is required to perform trend analysis up to global scale. Ziemke et al. (2005) using the Convective Cloud Differential (CCD) method on Total Ozone Mapping Spectrometer (TOMS) version 8 data from 1979 to 2003, found a statistically significant positive linear trend in the mid-latitudes but not in the tropics. Beig and Singh (2007) using the same data found an increasing trend of 7–9 % decade$^{-1}$ over some parts of south Asia, 4–6 % decade$^{-1}$ over the Bay of Bengal and 2–3 % decade$^{-1}$ over the central Atlantic ocean and central Africa up to 2005. Kulkarni et al. (2010) using Tropospheric Ozone Residual (TOR) data from TOMS, SAGE and SBUV instruments, calculated statistically significant trends over three Indian mega-cities during 1979–2005. They showed that ozone increased by 3.4 % decade$^{-1}$ in Delhi during monsoon period, while it increased by 3.4–4.7 % decade$^{-1}$ in Hyderabad and 5–7.8 % decade$^{-1}$ in Bangalore during pre-monsoon and post-monsoon, respectively. Ebojie et al. (2016) using the full record of SCIAMACHY limb-nadir matching data (2002-2011) retrieved regional and global tropospheric ozone trends. An insignificant positive trend in the order of 0.5 DU decade$^{-1}$ was noticed for the northern tropics (0-20°N) and in the order of 0.3 DU decade$^{-1}$ in the southern tropics (0–20°S). Regionally, they reported statistically significant trends of -1.6 % year$^{-1}$year$^{-1}$ over Northern South America (0–10°S, 75-45°W), of 1.6 % year$^{-1}$ in Southern Africa (5-15°S, 25-35°E), of 1.9 % year$^{-1}$ over Southeast Asia (15-35°N, 80-115°E), and a trend of 1.2 % year$^{-1}$ over Northern Australia (20-10°S, 100-130°E). Most recently, Heue et al. (2016) published a study about tropical tropospheric ozone trends using the CCD method on a harmonised dataset consisting of data retrieved from GOME, SCIAMACHY, GOME-2 and OMI satellite instruments from July 1995–December 2015 which are based upon different total ozone and cloud retrievals as well as merging approaches. The mean tropical tropospheric ozone trend that they found is 0.7 DU decade$^{-1}$ and regionally the trend reaches 1.8 DU decade$^{-1}$ near the African Atlantic coast, and -0.8 DU decade$^{-1}$ over

the western Pacific. Seasonally, they found that the trend over the South African coast maximises in summer, whereas the negative trend over the southwest Pacific ocean maximises during autumn.

The purpose of this study is to harmonise three individual tropical troposphere ozone (TTCO) datasets retrieved with the CCD method (Leventidou et al., 2016), and to investigate the tropical tropospheric ozone trend from the merged dataset between 1996 and 2015. This paper is organized as follows: Sect. 2 presents various scenarios used in order to harmonise the three separate datasets into merged time-series. Sect. 3 describes the regression model used to derive trends and presents the trend results (on global, regional, and local scale), along with their sensitivity to different harmonisation approaches used, and finally Sect.4 summarizes and discusses the findings of this study.

## 2 Harmonisation of the TTCO dataset

### 2.1 Tropical tropospheric $O_3$ data

Monthly mean TTCO data have been retrieved in a previous work of Leventidou et al. (2016) using the Convective Clouds Differential (CCD) method on GOME (Burrows et al., 1999), SCIAMACHY (Burrows et al., 1995; Bovensmann et al., 1999), and GOME-2 (Callies et al., 2000) total ozone and cloud data from 1996 to 2015. These instruments have different properties such as spatial resolution, cloud algorithms, overpass time, etc. The individual TTCO datasets have been created taking into account these specific characteristics and have been separately validated with integrated (until 200 $\mathrm{hPa}$) tropospheric ozone columns by ozonesondes from the SHADOZ network (Thompson et al., 2003) (see: Leventidou et al. (2016)). The biases between them have been found to be within -1.6 – 6.4 $\mathrm{DU}$ and the root mean square (RMS) deviation less than 13 $\mathrm{DU}$ for all the instruments. The uncertainty of the tropospheric ozone column retrieval with the CCD method is in the order of 3 $\mathrm{DU}$ ($\sim 10\%$). For most of the stations, the bias with the ozonesondes is within the retrieval uncertainty, with the exception of GOME-2 TTCO which is in the order of 5 $\mathrm{DU}$. Finally, the CCD TTCO from SCIAMACHY data have been compared with the Limb-Nadir-Matching (LNM) tropospheric $O_3$ columns up to 200 $\mathrm{hPa}$ altitude from the same satellite instrument, showing that the bias and the RMS values are within the ones calculated for the comparison with ozonesondes.

### 2.2 Correction offsets between GOME and GOME-2 with SCIAMACHY TTCO

In order to remove the biases between the instruments and create one consistent tropical tropospheric columns dataset from the CCD method for the whole timespan of the European satellites operation (1996–2015), correction offsets have been calculated. SCIAMACHY TTCO were used as reference for the correction offset calculation, since is the only instrument that overlaps (2002-2012) both with GOME and GOME-2 and has the smallest bias with respect to the ozonesondes (< 2 $\mathrm{DU}$). The average difference (bias) for each grid-box during the common years of the instruments operation (2002 for SCIAMACHY–GOME and 2007-2012 for SCIAMACHY–GOME-2) was computed and applied (added) to GOME and GOME-2 TTCO data. The mean biases, shown in Fig. 1, range between -6 and 6 $\mathrm{DU}$ for GOME, with positive differences (3–6 $\mathrm{DU}$) located mainly over land. There are also two stripes with positive biases appearing north of 7.5°N until 20°N, and between -5 and -7.5°S. For

GOME-2, the bias ranges between -8 and 0 DU, with differences getting smaller over land, especially over south America and north/central Africa. Possible reasons for the biases are the different cloud algorithms used for each instrument (SACURA for SCIAMACHY and FRESCO for GOME and GOME-2) and the small biases noticed in the total ozone columns (e.g. ∼ -2.5 DU between SCIAMACHY and GOME-2).

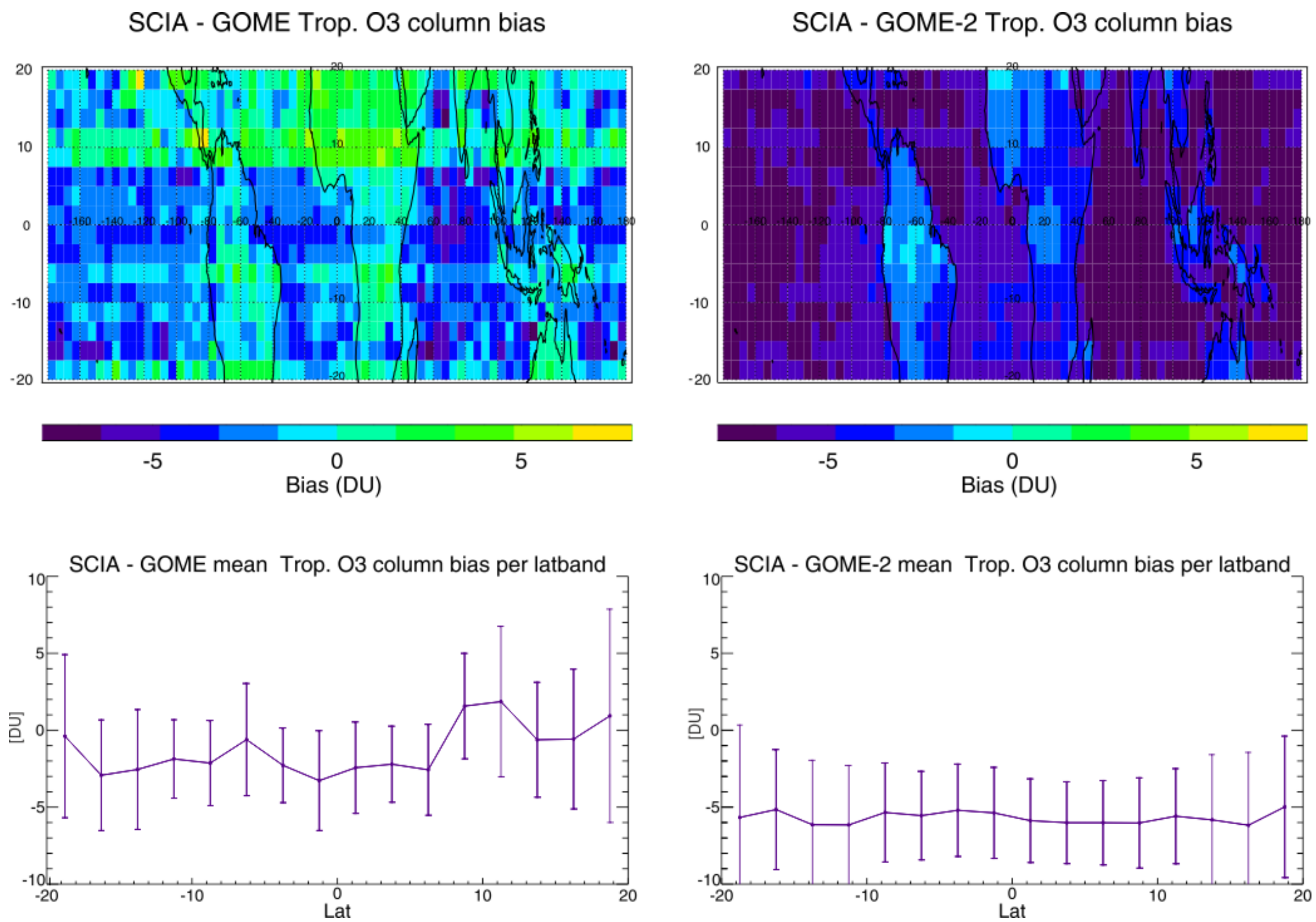

**Figure 1.** Correction offsets using SCIAMACHY TTCO as reference. (left) Correction offset for GOME: average difference of GOME from SCIAMACHY TTCO for the years 2002-2003. (right). Correction offset for GOME-2: average difference of GOME-2 from SCIAMACHY TTCO (in DU) for the years 2007-2012. The error bars denote the $1\sigma$ standard deviations of the latitudinally averaged biases.

The latitudinal dependence of the mean bias is shown at the bottom of Fig. 1. The average differences between GOME and GOME-2 with SCIAMACHY are generally negative (less than 5 DU) in all latitude bands with the exception of the northern tropical latitudes where GOME mean biases are positive (0–2 DU). GOME mean biases have stronger latitudinal variability than GOME-2. This behaviour may be explained by the short time of common operation (Jan. 2002–Jun 2003) between GOME and SCIAMACHY instruments. The $1\sigma$ standard deviation (uncertainty bars) of the mean bias per latitude band is comparable to the magnitude of the biases, ranging from less than 5 DU close to the equator to 7 DU for latitude bands close to the tropical

borders. For the case of GOME, the mean correction offset is -1.2 DU, whereas for GOME-2, it is -5.7 DU. The mean offset of GOME-2 is almost twice the CCD retrieval uncertainty (∼3 DU). For this reason and because of the large biases with the ozonesonde data, seems reasonable to apply a correction for the GOME-2 TTCO dataset.

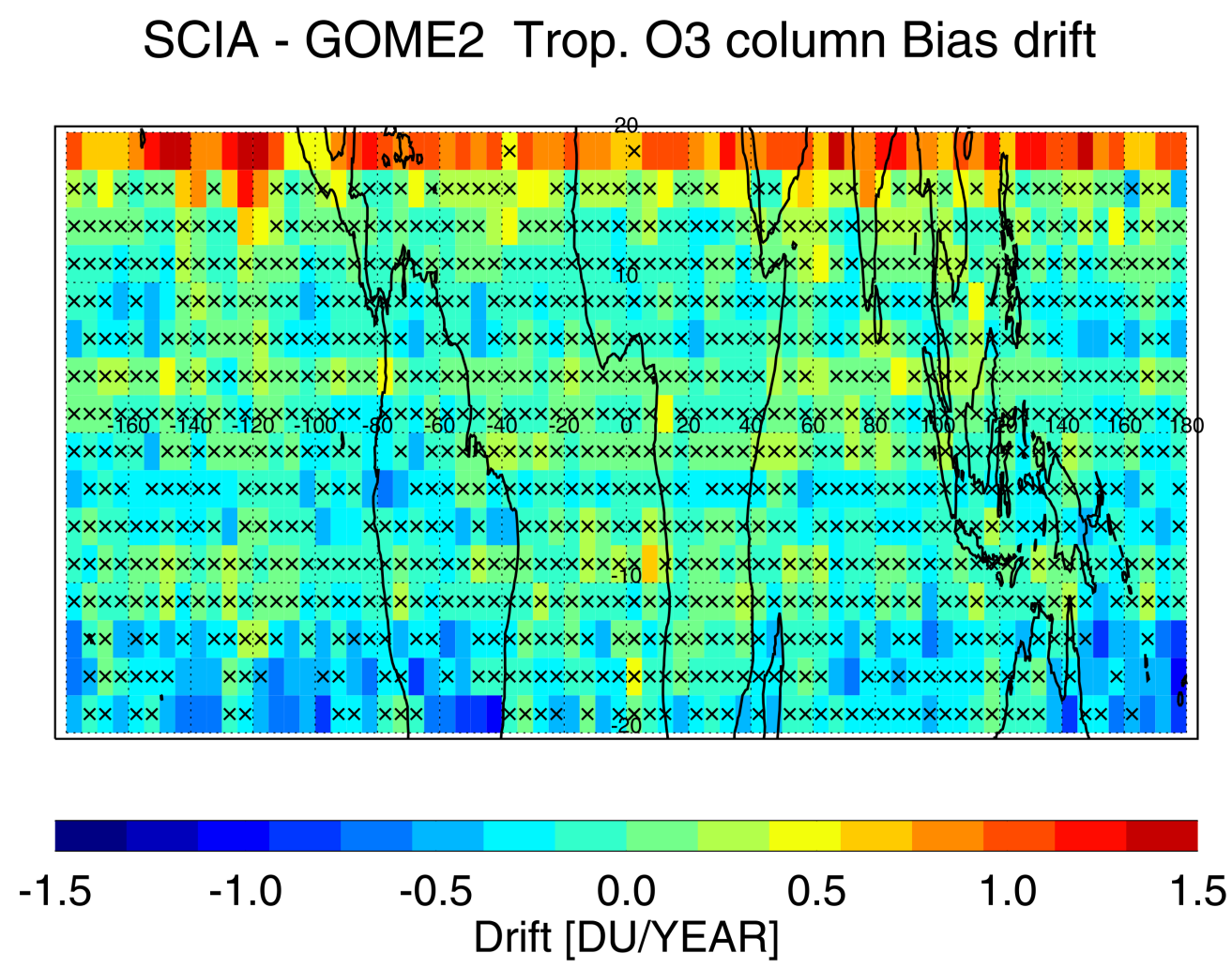

**Figure 2.** Trend in the correction offset for GOME-2. Black "x" denotes statistically non significant trend.

The drift on the average differences (bias), $\beta$, has been estimated using a simple linear regression model such as: Y = $\alpha$ + $\beta\cdot$ $X_t$, where Y is the time-series of the biases, $X_t$ is the time variable in months, and $\alpha$ is the offset. The drift between SCIAMACHY and GOME-2 (there are not enough overlapping years to calculate a trend in the GOME-SCIAMACHY difference time-series) is shown in Fig. 2. The drift is generally less than ∼0.4 DU per year and is statistically not significant ($\beta/\sigma_\beta$<2 (Weatherhead et al., 1998; Wilks, 2011)) for nearly all grid boxes, with the exception of the 17.5–20 $^o$N latitude band, where it is statistically significant and exceeds 1 DU year$^{-1}$. During local winter months at the tropical borders, there are often missing TTCO data owing to the movement of the ITCZ and the inability to retrieve a reliable stratospheric $O_3$ column. For this reason, calculated drifts for these latitudes are not reliable despite the fact that they might appear to be statistically significant. They do not cover the entire studied period (2007–2012) but only a subset of them. Consequently, the trend of the correction offsets is not considered here.

### 2.3 Six Harmonisation scenarios

The creation of a consistent tropical tropospheric ozone column dataset from multiple satellite instruments demands a careful selection of the appropriate harmonisation approach, since it introduces additional uncertainty in the merged dataset. For this purpose, six harmonisation scenarios have been tested all of them using the SCIAMACHY TTCO dataset as a reference, which is in the middle of the time period, as follows:

- **Scenario 1**: No correction applied to GOME data (which maybe justified by the very short overlap period), while GOME-2 is corrected using for each grid-box the mean bias with respect SCIAMACHY for the common years of operation (2007–2012 for GOME-2).
- **Scenario 2**: No correction applied to GOME data and the average bias (-5.7 DU) with respect SCIAMACHY is added to all GOME-2 TTCO data.
- **Scenario 3**: GOME and GOME-2 have been corrected using for each grid-box the mean bias with respect to SCIAMACHY for the common years of operation.
- **Scenario 4**: The average bias with respect to SCIAMACHY (-1.2 DU) is added to all GOME TTCO data, whereas GOME-2 TTCO has been corrected using for each grid-box the mean bias with respect to SCIAMACHY for the common years of operation (2002 for GOME and 2007-2012 for GOME-2).
- **Scenario 5**: The average bias with respect to SCIAMACHY (-1.2 DU) for GOME and for GOME-2 (-5.7 DU) is added to all GOME and GOME-2 TTCO data respectively.
- **Scenario 6**: No correction applied to GOME, whereas for GOME-2 both the bias and the drift is included in the correction of GOME-2 TTCO in each grid-box.

After the correction terms for all scenarios have been applied to the original data, the "corrected" GOME (1996-2002) and GOME-2 (2007-2015) TTCO were averaged with the ones from SCIAMACHY (2003-2012) for the overlapping months (Jan. 2002–Jun. 2003 and Jan. 2007–Dec. 2012, respectively).

**Table 1.** Mean differences (in DU) between the harmonised TTCO datasets using six different harmonisation scenarios with integrated ozone columns until 200 hPa from nine ozonesonde stations. The regions where the harmonisation scenarios have the smallest biases with ozonesondes are marked bold.

| CCD – Sondes TTCO [DU] /Site | scenario 1 | scenario 2 | scenario 3 | scenario 4 | scenario 5 | scenario 6 |
|---|---|---|---|---|---|---|
| **Ascension (8S,14.4W)** | **0.03** | -0.14 | -0.77 | -0.42 | -0.60 | **0.03** |
| **Paramaribo (5.8N,55.2W)** | **-1.21** | -2.28 | -1.28 | -1.44 | -2.52 | **-1.21** |
| **Java (7.6S,111E)** | **-0.11** | -0.12 | -1.12 | -0.54 | -0.55 | **-0.11** |
| **Natal (5.4S,35.4W)** | 0.56 | 0.63 | **-0.21** | 0.22 | 0.28 | 0.57 |
| **Samoa (14.4S,170.6W)** | -0.25 | **0.09** | -1.35 | -0.61 | -0.23 | -0.28 |
| **Nairobi (1.3S,36.8E)** | 1.81 | 1.10 | 1.80 | 1.48 | **0.74** | 1.84 |
| **Kuala Lumpur (2.7N,101.7E)** | -1.81 | -2.12 | -2.12 | -2.14 | -2.48 | **-1.78** |
| **Hilo (19.4N,155.4W)** | 0.67 | 0.65 | **0.17** | 0.19 | 0.23 | 0.88 |
| **Fiji (18.1S,178.4E)** | 0.19 | **-0.09** | -0.58 | -0.21 | -0.45 | -0.55 |

In order to conclude which is the most suitable harmonisation scenario, the various merged datasets were compared with integrated ozone columns up to 200 hPa altitude from nine ozonesonde stations: (a) Ascension (8°S, 14.4°W), b) Paramaribo (5.8°N, 55.2°W), c) Java (7.6°S, 111°E), d) Natal (5.4°S, 35.4°W), e) Samoa (14.4°S, 170.6°W), f) Nairobi (1.4°S, 36.8°E), g) Kuala Lumpur (2.7°S, 101.7°E), h) Hilo (19.4°N, 155.4°W), and (i) Fiji (18.1°S, 178.4°E)). As seen in Table 1, the mean bias between the six harmonised TTCO datasets and the ozonesondes range between -2.5 and 1.8 DU which is well within the retrieval uncertainty. However, the biases of each scenario with ozonesondes are very close to each other for every station. The same, occurs for the correlation between the harmonised TTCO datasets and the ozonesondes (not shown here). Although the comparison between the TTCO from the individual harmonised scenarios and the ozonesonde data does not favor clearly any harmonisation scenario, the scenarios that can be confidently rejected according to this comparison are scenarios 3, 4 and 5, which have the biggest bias with the ozonesondes. Scenario 6 presents smaller bias at four out of nine ozonesonde stations, whereas scenario 1 at three out of nine ozonesonde stations. Nevertheless, scenario 6 has larger biases with respect to ozonesondes compared to scenario 1 (with the exception of one station). As shown earlier, the drift in the GOME-2 data (scenario 6) is statistically insignificant at most of the grid-boxes and as will be shown later introduces artifacts in tropospheric trends. For these reasons, scenario 1 has been selected to be the preferred harmonisation scenario for merging the TTCO datasets. All results (without explicit indication of the harmonisation scenario used) presented here are based on harmonisation scenario 1.

## 3 Tropical tropospheric ozone trends

Changes in ozone precursor emissions due to urbanization and land use, along with changes in the atmospheric oscillations which affect processes that modulate the tropical upwelling or the horizontal ozone transport, may cause long-term changes in the tropospheric ozone burden and influence the photochemical ozone production and loss in the troposphere (Ziemke and Chandra, 2003; Solomon et el., 2007; Chandra et al., 2009; Voulgarakis et al., 2010; WMO, 2011; Neu et al., 2014; Monks et al., 2015). Some of these factors can be represented by periodic seasonal proxies, such as the El Niño Southern Oscillation (ENSO), the quasi-biennial oscillation (QBO), and the solar cycle (SC). These indexes are embodied in the trend model described in Subsection 3.1. The seasonal variation of the linear trend is also included using harmonic functions which represent the annual, semi-annual and quarterly harmonic oscillations.

### 3.1 The multi-linear regression trend model

The time series of the monthly mean tropical tropospheric ozone columns $Y_t$ at a specific latitude and longitude *(i,j)* (running every 2.5° and 5°, respectively) can be generally described by the following trend model:

$$Y_t(i,j) = \alpha(i,j) + \beta(i,j) \cdot X_t + S_t(i,j) + R_t(i,j) + N_t(i,j) \qquad (1)$$

where $a$ is the offset TTCO for the first month t=1, $\beta$ the linear trend in DU month$^{-1}$, X the time variable (months running from zero to 239) covering the years 1996–2015, $S_t$ is the seasonal variation, $R_t$ are the terms with the various proxies (ENSO, QBO, solar cycle) and $N_t$ is the noise of the time series, representing the unexplained portion of the variability in the fit. Analytically the seasonal cycle is modeled by a Fourier series (see Eq. 2), with $\gamma_{11}$, $\gamma_{21}$, $\gamma_{12}$, $\gamma_{22}$, $\gamma_{13}$, $\gamma_{23}$ being the regression coefficients for 12-, 6- and 4-month periodicities, with sine and cosine terms for each periodicity, respectively.

$$S_t(i,j) = \sum_{n=1}^{3} (\gamma_{1_n} \cdot sin(\frac{2 \cdot \pi \cdot n \cdot t}{12}) + \gamma_{2_n} \cdot cos(\frac{2 \cdot \pi \cdot n \cdot t}{12})) \tag{2}$$

$R_t$, represents the time dependent regression coefficients for the ENSO, QBO, and solar cycle proxies which can be expressed as:

$$R_t = \delta \cdot ENSO_t + \varepsilon \cdot QBO_{30_t} + \zeta \cdot QBO_{50_t} + \eta \cdot SC_t \tag{3}$$

Since the tropospheric ozone lifetime approaches a month, the pattern of tropospheric ozone for a month has the tendency to recur on the next month. Even after removing to the largest extent the seasonal and other effects in the time series shown in Eq. 1, there is still a month-to-month correlation ($\phi$) in residuals. This phenomena is called persistence (Wilks, 2011) and is quantified by the degree of autocorrelation of a parameter, shifted by p time steps (lag p). Therefore, the autocorrelation of the noise is included in the model as explained by Weatherhead et al. (1998).

### 3.2 Sensitivity and uncertainty of the trend

The multivariate linear regression model (Eq. 1) and correction for AR(1) have been applied to six individual harmonised CCD tropical tropospheric ozone columns datasets (see: Section 2) from 1996 to 2015. Figure 3 shows the trend map and its statistical significance for the six scenarios. The tropospheric $O_3$ trends from all scenarios range between ∼-4 and 4 DU decade$^{-1}$, with mean values between 0 and 0.8 DU decade$^{-1}$ without any of them being statistically significant. The maximum trend difference among all six harmonisation scenarios is on average 2 DU decade$^{-1}$ exceeding the $2\sigma_\beta$ uncertainty of the trends which is ∼ 1.2 DU decade$^{-1}$. These differences on the trends among the differently harmonised datasets reveal the additional uncertainty which is inherited to the trend from the harmonisation procedure of multiple TTCO datasets. The maximum absolute differences (>3–6 DU decade$^{-1}$) are found mainly over land and more specifically over south America, and northern Africa, while the minimum absolute differences are over the Oceans with the exception of the Indian and the southern Pacific Ocean. Nevertheless, all scenarios shown in Fig.3 agree that there is a positive trend of tropospheric ozone over the south tropical Atlantic Ocean, and some parts of central Africa and India, while a negative trend appears over the Caribbean sea, the north and south Pacific Ocean.

Scenarios 1, 4, and 6 have a similar pattern with each other which is caused by the absence of correction of the GOME TTCO dataset. Nevertheless, the range of the trends is different, with scenario 4 showing higher positive trends (∼ 2 – 4 DU decade$^{-1}$), mainly over Africa, south America and the southern tropical borders. Scenarios 2 and 5 have also similar pattern with each other, driven by the average offset applied to GOME-2 data. The pattern of these scenarios consist of a characteristic decrease

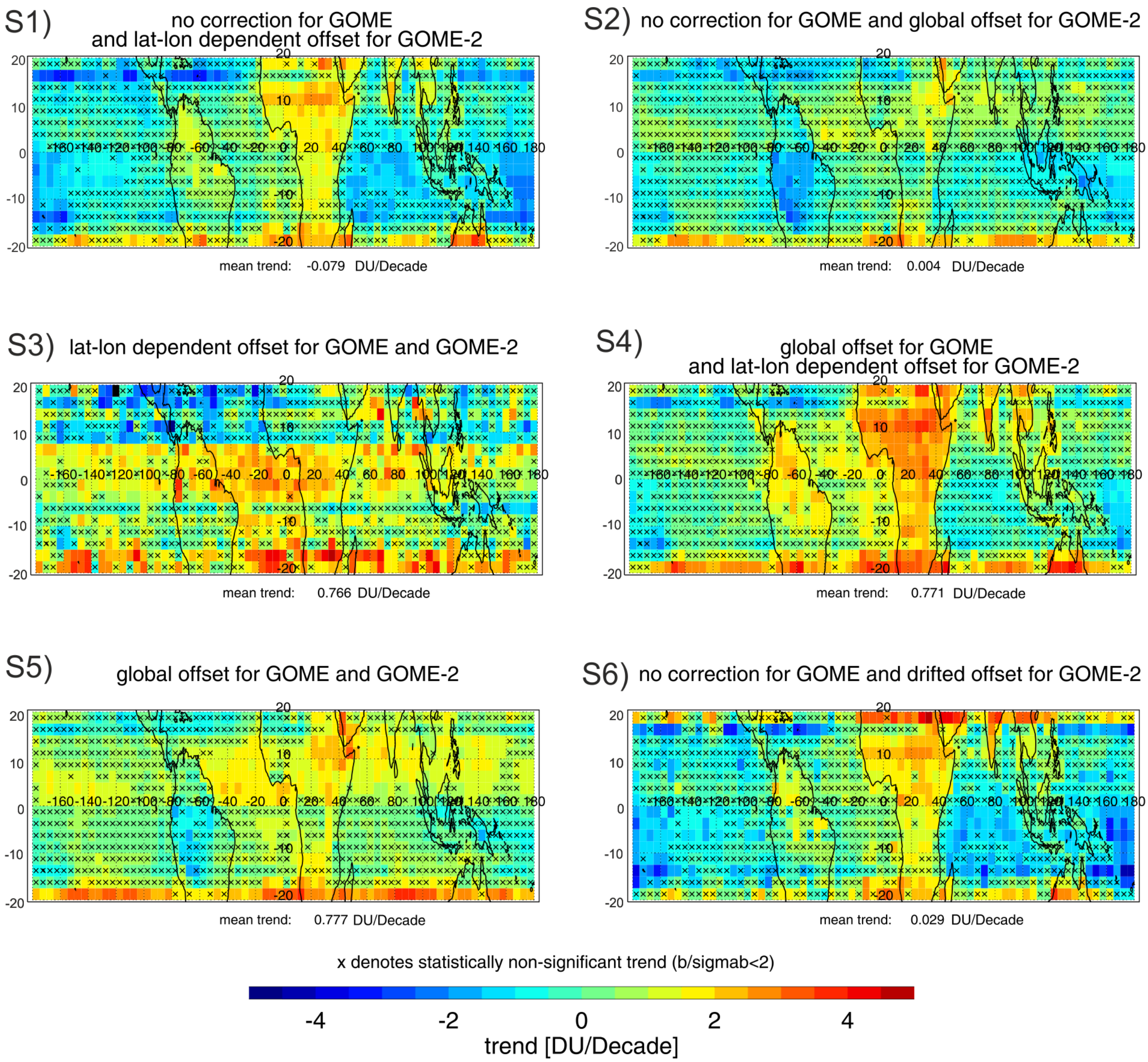

**Figure 3.** Tropical tropospheric ozone trends using a linear multivariate first order auto-regression model for 6 harmonisation scenarios, see Sec. 2.3. The trends are given in DU per decade. Grid-boxes marked with "x" are statistically non-significant at the 95% confidence level.

in tropospheric ozone (~ -2 DU decade$^{-1}$) over central-south America and over the Indonesian peninsula. The tropospheric $O_3$ trends calculated with scenario 3 repeat the meridional pattern of GOME correction offsets (see: Fig. 1), which appears as an artifact in the trend results.

### 3.3 Tropical tropospheric ozone trend results

From now on, the discussion about tropical tropospheric ozone trend refers to the preferred harmonisation scenario (scenario 1). As shown in Fig. 3, panel S1, the TTCO trend varies between -3.2 and 3.7 DU decade$^{-1}$, and the average trend for the period 1996–2015 is statistically non-significant and equal to -0.08± 1.23 DU decade$^{-1}$ ($2\sigma$). The noise is random (white noise) following very well a Gaussian distribution. The multivariate regression model (Eq. 1) has been also applied to the

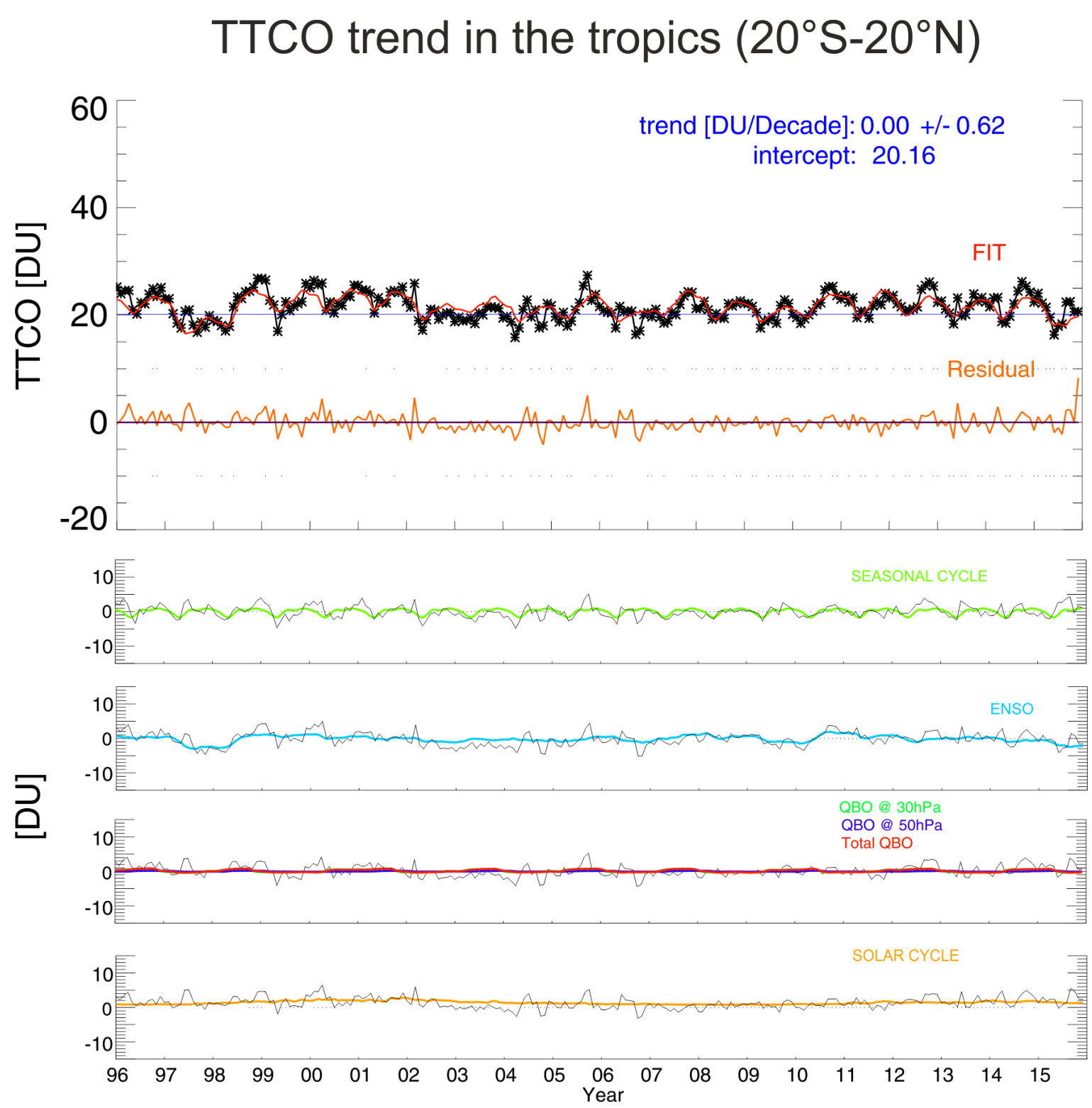

**Figure 4.** Tropospheric ozone trends for the global tropical tropospheric ozone for the period 1996 to 2015. Top: The multivariate linear trend (black), the fit (red) and the residual (orange) are over-plotted. The global tropical tropospheric ozone trend is equal to zero and the $2\sigma$ uncertainty of the trend is ±0.62 DU decade$^{-1}$. The next panels show the harmonic functions (green), ENSO (light blue), QBO (red), solar (orange). Overlaid in black for all proxies are the time series with all fit terms removed except the particular fit parameter.

global tropical mean time-series (20$^o$S–20$^o$N). The fit results are shown in Fig. 4. The global tropical trend equals 0.0±0.6 DU decade$^{-1}$. This means that there is no significant trend for tropospheric ozone in the tropics. This result is in agreement

with Ziemke et al. (2005) and Ebojie et al. (2016) who found no significant global tropospheric ozone trends in the tropics. The tropical mean tropospheric ozone time-series (black stars) shows a seasonal cycle with higher values in late summer-autumn months. The time-series are well followed by the regressed tropospheric ozone (red line) and the residual (orange line in upper panel) is less than 3 DU. The seasonal cycle contributes the most to the TTCO variability in the tropics by about ±3 DU. Tropical tropospheric ozone is reduced by -3 DU during El Niño years (1997-98, 2006-07, 2009-10, 2015) and slightly increases by 1 – 2 DU during strong La Niña years (1999-00, 2007-08, 2010-11). QBO and the solar cycle, practically do not contribute to the inter-annual mean tropical tropospheric ozone variability. Overlaid in black for all proxies are the time series with all fit terms removed except the particular fit parameter. This allows us to relate the magnitude of changes due to a certain process to the observed residuals (or unexplained variations).

Figure 5 summaries the tropical tropospheric ozone trends calculated in a $2.5^o \times 5^o$ grid as derived from the merged CCD TTCO dataset between 1996 and 2015. Fig. 5b shows the $2\sigma$ of the trend, which is in the order of ∼0–4 DU decade$^{-1}$ (mean: 1.2 DU decade$^{-1}$), with greater values at the tropical borders and values close to zero along the equator. Fig. 5c shows the correlation between the model and the time-series. The correlation coefficient reaches 1 over the north and central-east Pacific and the southern Atlantic Ocean. The regions of smaller correlations are mostly over the west Pacific, the Caribbean sea, the south-east Asia, and over the central African continent. The main reason for the low correlation is the very weak seasonal cycle observed in these regions. Fig. 5d shows the RMS between the time-series and the model fit. The RMS is less than 3 DU close to equator and reaches 7 DU at the tropical borders. Fig. 5e presents only those grid boxes where the trend is statistically significant and exceeds the maximum difference of the trends calculated from all six scenarios. This additional criterion (to exceed the differences between harmonisation scenario) allows us to identify grid boxes that have significant trends with higher confidence. Following this stronger criterion, tropospheric ozone increases over some parts of central Africa (∼2 DU decade$^{-1}$), southern Africa and Atlantic Ocean (∼2 – 3 DU decade$^{-1}$), India (∼2 DU decade$^{-1}$) and Oceania (∼3 – 4 DU decade$^{-1}$) and decreases over the Caribbean sea and parts of North Pacific Ocean (∼-2 – -3 DU decade$^{-1}$), as well as over some regions of the southern Pacific Ocean (∼-2 DU decade$^{-1}$) seem to be relevant, however, for all other grid boxes trends are highly uncertain and mainly dependent on the choice of the harmonisation scenario. The negative trends appearing as a stripe at northern latitudes (Caribbean sea and northern Pacific) may still be an artifact of the data-set (low sampling of data). Finally, Fig. 5f shows the tropical tropospheric ozone trends in per cent per year (% year$^{-1}$) that are statistically significant for the TTCO data harmonised according to scenario 1 (S1). Here the maximum increase is noticed over central Africa, ∼3% year$^{-1}$, over southern Africa, south tropical Atlantic and Oceania ∼1.5% year$^{-1}$, and finally over India and south-east Asia ∼1% year$^{-1}$. The maximum tropospheric ozone decrease is noticed over the Carribean sea and the noth-east tropical Pacific, about ∼-2% year$^{-1}$, followed by the central-south Pacific and Indian Ocean, ∼-1% year$^{-1}$.

### 3.3.1 Regional and Mega-cities tropical tropospheric ozone trends

We also studied regional trends focusing on the regions where the trends are statistically significant. The TTCO have been regionally averaged for eight regions and the regression analysis applied to them. The regions are: A: Caribbean Sea (15° – 17.5°,-85° – -45°), B: India(10° – 20°, 70° – 85°), C: north-south America (0° – 10°, -75° – -60°), D: North Africa (5° – 15°,

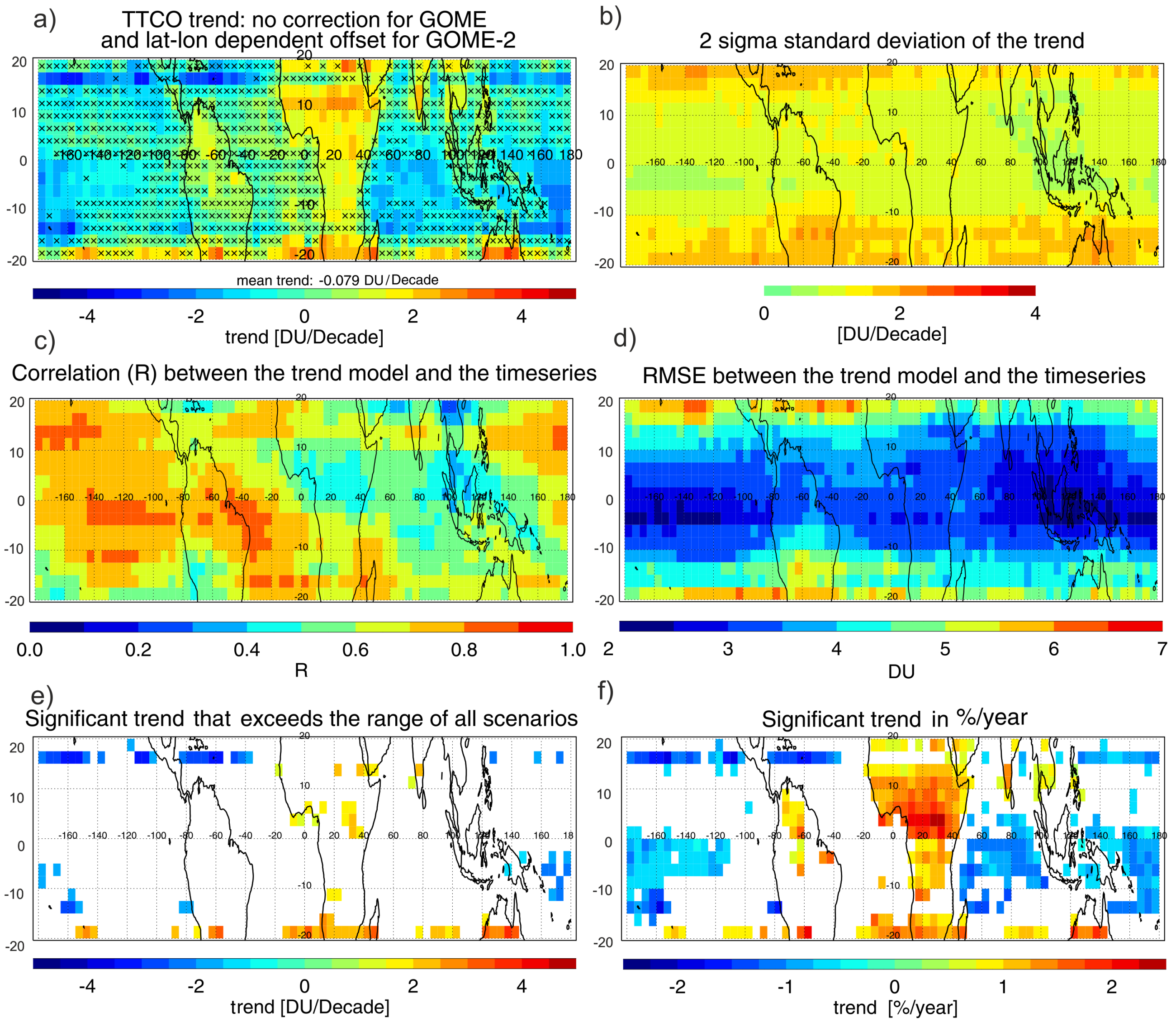

**Figure 5.** (a) Tropical tropospheric ozone trends using a linear multivariate first order auto-regression model for the selected harmonised scenario 1 in DU decade$^{-1}$. Grid- boxes marked with "x" are statistically non-significant at the 95% confidence level (b>2$\sigma_\beta$). b) 2$\sigma$ standard deviation of the trend. c) The correlation coefficient, R, between the multi-linear trend model fit and the original time-series. d) The RMS error between the trend model and the time-series. e) The statistically significant trend that exceeds the maximum absolute difference of the trends calculated for all six scenarios. f) The significant tropical tropospheric ozone trend in % year$^{-1}$.

-17.5° – 50°), E: east Pacific Ocean (0° – 7.5°, -180° – -110°), F: Indian Ocean (0° – 7.5°, 50° – 100°), G: west Pacific Ocean (0° – 7.5°, 160° – 180°), and H: southern Africa (-20° – -12.5°, 10° – 50°).

As shown in Figure 6 and Table 2, regions B, C, D and H show significant increase in the order of 1–1.5 DU decade$^{-1}$ and regions A, E, F, and G a significant ozone decrease in the order of -1.2–1.9 DU decade$^{-1}$.

**Table 2.** Regional tropospheric ozone trends in 8 tropical regions. Bold are the regions where the trend is greater than three times the standard deviation of the trend ($3\sigma$).

| **Area** | **Tropospheric O3 trend $\pm 2\sigma$ in DU decade$^{-1}$** |
|---|---|
| A) Caribbean sea | -1.59 ± 1.30 |
| B) India | 1.10 ± 0.86 |
| C) North South America | 0.99 ± 0.94 |
| D) North Africa | 1.54 ± 1.09 |
| **E) East Pacific Ocean** | **-1.21 ± 0.65** |
| **F) Indian Ocean** | **-1.61 ± 0.83** |
| **G) West Pacific Ocean** | **-1.87 ± 0.72** |
| H) South Africa | 1.44 ± 1.28 |

The observed significant positive changes in tropospheric $O_3$ over north Africa and parts of the Arabian sea (D), south Africa and the southern African outflow (H), parts of India (B), and north south America (C) agree well with results of Lelieveld et al. (2004), Beig and Singh (2007), Kulkarni et al. (2010), Ebojie et al. (2016) and Heue et al. (2016) who also noticed an increasing ozone trend over these regions. They can be attributed to changes in anthropogenic $NO_x$ and other tropospheric $O_3$ precursors, due to population and energy consumption increases, which are transported to these areas (Hilboll et al., 2013b; Kulkarni et al., 2010; Dahlmann et al., 2011; Schneider et al., 2015; Cooper et al., 2014; Duncan et al., 2016; Hilboll et al., 2017). Biomass burning may also have an impact on tropospheric ozone changes. For example, the burned area in southern tropical Africa increased by 1.8 %/yr during the period 2000 to 2011 (Giglio et al., 2013). Ziemke et al. (2009b) and Wai et al. (2014) estimated that biomass burning can contribute to an increase in tropospheric ozone column by $\sim$20%. Additionally, changes in meteorology, convection, and dynamical oscillations, such as the MJO, stratospheric intrusions (STE) and shorter timescale atmospheric dynamics or cyclones may have influence the transport of pollutants and contribute locally to observed tropospheric ozone changes (Ziemke et al., 2009b; Beig and Singh, 2007; Parrish et al., 2009; Ebojie et al., 2016; Chandra et al., 2004; Oltmans et al., 2004; Sauvage et al., 2007). Another factor that mayinfluence tropospheric ozone are the changes in stratospheric ozone column. For example, an increase in the tropical upwelling caused by a stronger Brewer-Dobson circulation is expected to reduce both lower stratospheric and the total column ozone in the tropics, increasing the UV-B radiation reaching

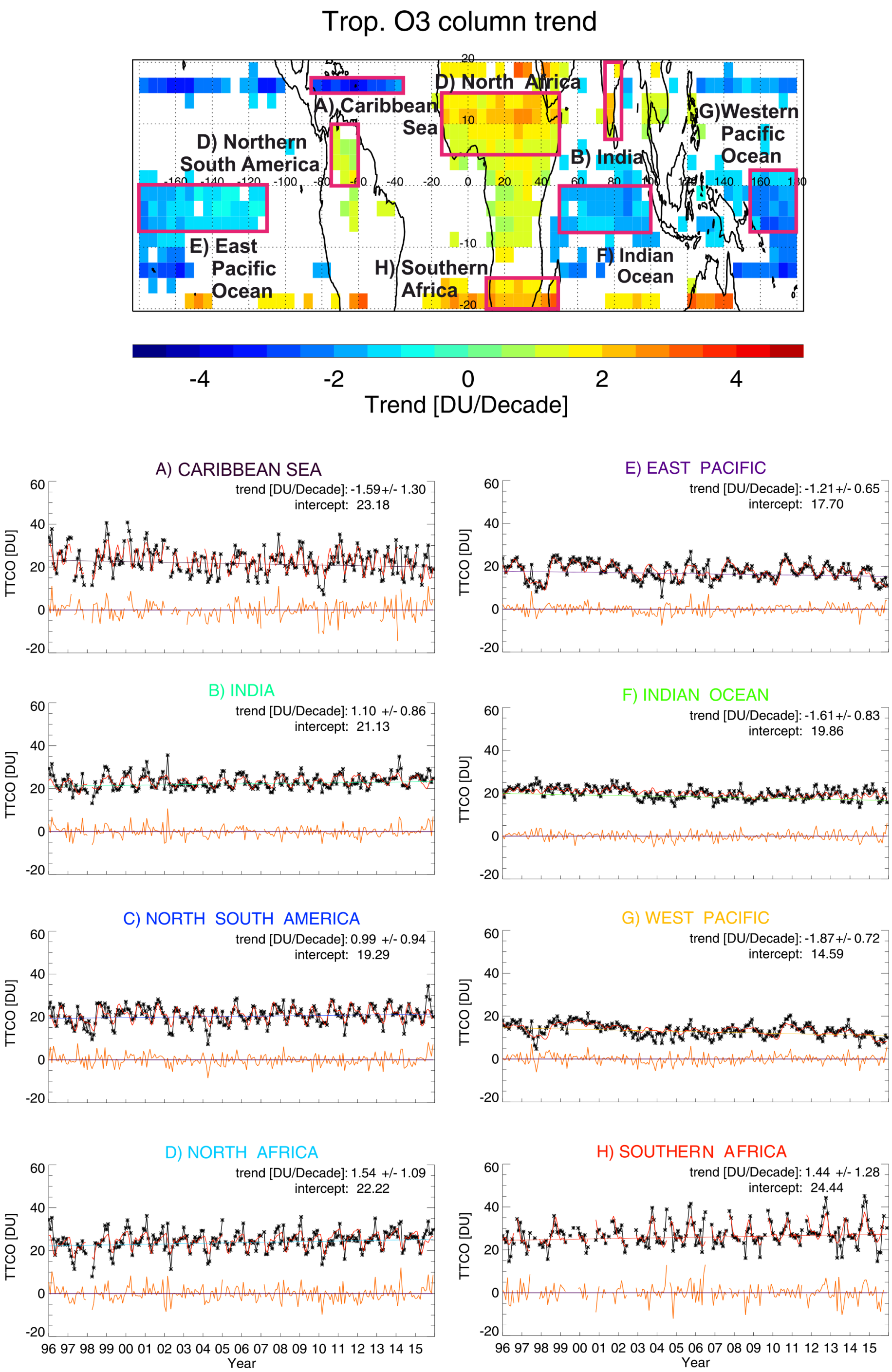

**Figure 6.** Tropical tropospheric ozone trend in A) central America, B) India, C) east Pacific Ocean, D) South America, E) central Atlantic Ocean, F) Indian Ocean, G) south Atlantic Ocean, and H) southern Africa.

the troposphere (WMO, 2014). This could result in an enhance of tropospheric ozone photolysis (photochemical ozone sink). However, the increase of UV-B radiation at the surface would also lead to increased concentrations of OH (hydroxyl radicals) and subsequently increased concentrations of $HO_2$ and $RO_2$ radicals, which may enhance the production of ozone if $NO_x$ are available (e.g. in maga-cities) (UNEP, 1998). Consequently, there are multiple feedbacks from these changes that could either increase or decrease ozone in the troposphere.

The negative changes in TTCO over the Caribbean sea (A) are in agreement with the results of Ebojie et al. (2016). Although they might be influenced by the decrease in $NO_x$ emissions over the north American continent (Duncan et al., 2016; Hilboll et al., 2013b) or by changes in stratospheric intrusions via the tropopause foldings (Hwang, et al.; Ojha et al., 2017), the observed trends over the northern and southern tropical latitudes (>18$^o$N and S) should be generally interpreted with caution since they are influenced by low sampling of data. Despite the fact that might appear to be statistically significant, they should be interpreted with caution since they are influenced by gaps in the TTCO time series due to the movement of the ITCZ, which reduces the cloudy data during local winters and makes the above cloud ozone column (ACCO) retrieval difficult, violating in some cases the invariance of the ACCO per latitude band.

The decreasing trend over the Pacific (E and G) and Indian (F) Oceans agrees well with Heue et al. (2016). It might be associated with changes in the burden of organic and inorganic halogens on these areas as well as changes in dissolved organic matter (DOM) photochemistry in surface waters could be an additional source of volatile organic compounds that can contribute to ozone destruction (Dickerson et al., 1999; Ebojie et al., 2016). Additionally it may be attributed to changes in the humidity burden of the troposphere. For example, Fontaine et al. (2011) indicated that the Outgoing Long-wave Radiation (OLR), which allows to differentiate between clear-sky (high OLR) and deep convective regions (low OLR) has been decreasing over these regions, which can indicate deeper convective clouds appearing over the Caribbean, the west-central Africa in summer and the Indian Ocean in autumn. The increased deep convection is associated with ozone loss due to convective outflow and increased cloudiness and humidity which contribute to photochemical $O_3$ loss (Morris et al., 2010; Wai et al., 2014). Fontaine et al. (2011) showed that the location of OLR minima has been shifted northwards which can be associated with a shift on the ITCZ by 0.5 – 0.8 $^o$ northwards. These changes are subsequently associated with changes in the location of tropical jets, with changes in rainfall amounts and weather systems. All these changes maybe responsible in some degree for the statistically significant tropospheric ozone trends observed close to the location and the branches of the ITCZ (e.g in the Indian and Pacific Oceans), but their contribution remains vague.

#### 3.3.2 Seasonal tropospheric $O_3$ trends

Seasonal tropospheric $O_3$ trends can be very useful for understanding the connection between the factors (e.g. meteorology or emissions) that contribute to tropospheric ozone changes and its distribution. For this reason, the multi-linear regression model has been applied to Dec.–Feb., Mar.–May, Jun.–Aug., and Sep.–Nov. TTCO time-series and proxies (ENSO, QBO, solar cycle) in order to calculate TTCO trends for winter, spring, summer and autumn respectively, with the only difference that the sine and cosine terms that reflect the seasonal cycle are neglected in the regression. According to Fig. 7, the maximum decreasing trends appear during winter over the northern tropical Atlantic and Pacific Oceans (~-4 DU decade$^{-1}$). These air masses are more

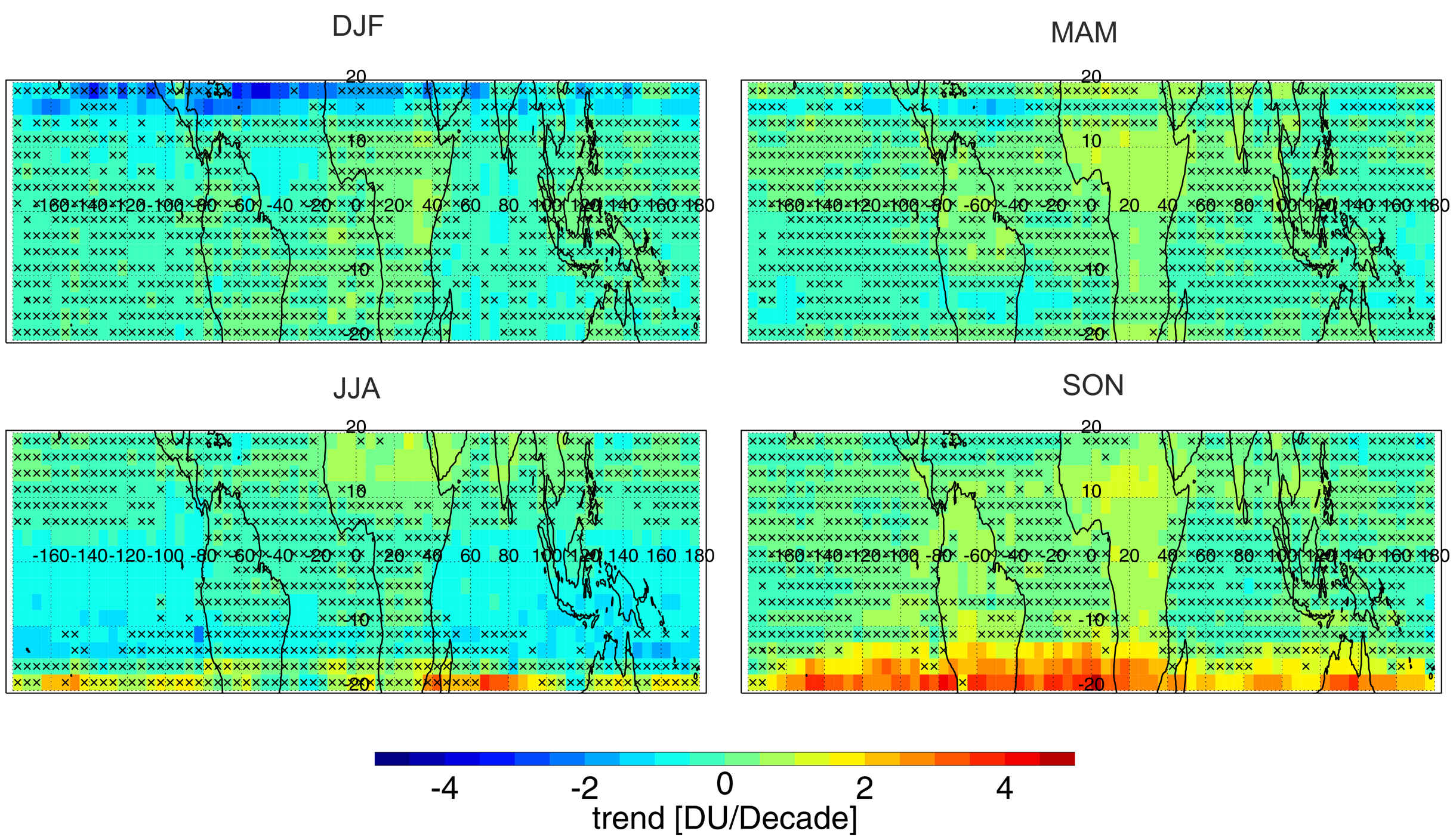

**Figure 7.** Tropical tropospheric ozone trends for winter (DJF), spring (MAM), summer (JJA), and autumn (SON) for the years 1996 to 2015.

affected by changes occurring in the mid-latitudes due to the southward movement of the ITCZ in winter and the strong westerly air flow over the tropical borders in winter (Ebojie et al., 2016). Therefore, it is assumed that changes in ozone precursors, such as $NO_2$, over North America and Europe may have affected the $O_3$ trends over these tropical latitudes (Logan et al., 2012; Hilboll et al., 2013b). The winter decrease might also be associated with the limited number of TTCO measurements on the northern tropical borders, thus it demands a more careful investigation. The trends are mostly insignificant during spring, with the exception of Africa where they are $\sim$ 1 DU decade$^{-1}$ and some parts over South America where ozone is decreasing by less than 1 DU decade$^{-1}$. During summer, ozone shows a slight statistically significant decrease over the Pacific and Indian Oceans (1-2 DU decade$^{-1}$). Possible reasons for tropospheric ozone decrease over the oceans may be related to changes in sea surface temperatures (SSTs) which are closely tied to the tropospheric humidity (Trenberth, 2011; IPCC, 2007). As discussed earlier, water vapor in the troposphere consists of one of the most important sinks of tropospheric ozone (Jacob, 2000). An increase in vertical convective patterns over the tropical oceans may result in lower ozone mixing ratios in the upper troposphere where the WFDOAS retrieval is more sensitive (Morris et al., 2010; Wai et al., 2014; Fontaine et al., 2011; Ziemke et al., 2008; Coldewey-Egbers et al., 2005). Several studies have shown that the total column of water vapour (TCWV) has increased over the tropics. Mieruch et al. (2014) and Trenberth et al. (2005) found that the TCWV has increased by $\sim$ 1–2 % decade$^{-1}$ over the oceans. Chen and Liu (2016) found that also the precipitable water vapor (PWV) increased by 1–2% in the tropics between 1992–2014. The precipitation increase is about 4% over the ocean, while a decrease of 2% is found over land in the latitude range 25$^o$S

to 25°N, between 1979 and 2001 (Adler et al., 2003). The significant positive trend of ozone at the southern tropical Atlantic, southern Africa, South America, and Oceania maximise during autumn (∼4 DU decade$^{-1}$). According to MODIS/TERRA Fire Radiative Power (mW/m$^2$) data (https://disc.gsfc.nasa.gov/neespi/data-holdings/mod14cm1.shtml) autumn is the season with the most intense fires over southern Africa and South America. Hence, it is very likely that biomass burning could be the origin of the observed ozone increase.

### 3.3.3 Tropospheric ozone trends over mega-cities

On local scales, the impact of anthropogenic emissions at mega-cities are of great interest since they affect human health of millions of people. Tropospheric ozone trends at grid-boxes ( 2.5°×5°) surrounding 10 tropical mega-cities have been selected in order to perform a quantitative comparison with other studies. The selected mega-cities in descending order of population are: Jakarta (-6 °S, 106.7 °E, 26 million people), Mexico (19.4°N, 99.1 °W, 24 million people), Manila (14°N, 120°E, 22 million people), Mumbai (19°N, 72°E, 21 million people), Bangkok (13.7°N, 100.5°E, 14 million people), Lagos (6°N, 3°E, 13 million people), Kinshasa (-4.4°S, 15.3°E, 10 million people), Bangalore (13°N, 77.6°E, 10 million people), Lima (-12.1°S, 77°W, 10 million people), and Nairobi (-1.3°S, 36.8 °E, 5 million people).

The trend results are presented in Table 3. The tropospheric ozone trends have been calculated with the regression model described in Section 3.1, using tropospheric $O_3$ data from our study (first two columns of Table 3). Additionally, tropospheric ozone data created by Heue et al. (2016) using the CCD method as well on GOME, SCIAMACHY, GOME-2 and OMI satellite measurements from July 1995–December 2015 (data taken from: http://www.esa-ozone-cci.org/?q=node/160) are presented in the third column of Table 3). The fourth column in the table shows the trends calculated by Ebojie et al. (2016) (in % year$^{-1}$) using the Limb/Nadir Matching technique on SCIAMACHY ozone data from 2002 to 2011, along with tropospheric $NO_2$ trends from Schneider et al. (2015) (fifth column) using SCIAMACHY (0.25°×0.25° degrees) $NO_2$ data, and Hilboll et al. (2013b) (sixth column) using multi-instrument (GOME, SCIAMACHY, GOME-2 and OMI gridded at 1°×0.5° degrees) $NO_2$ data (in molecules cm$^{-2}$ decade$^{-1}$).

Using our CCD data, statistically significant trends at the 95% confidence level ($|\beta| > 2 \cdot \sigma_\beta$) are found in Manila (1.1 ± 0.7 DU decade$^{-1}$), Bangkok (1.4 ± 0.9 DU decade$^{-1}$), and Kinshasa (1.3 ± 0.9 DU decade$^{-1}$). Elsewhere, the trend is less than 1 DU decade$^{-1}$ (< 0.5 % year$^{-1}$), while negative but insignificant trends of less than 0.5 DU decade$^{-1}$ are noticed in Jakarta (-0.2 ± 0.9 DU decade$^{-1}$), Mexico (-0.3 ± 0.9), and Lima (-0.5± 1.5 DU decade$^{-1}$). Using the Heue et al. (2016) dataset, statistically significant positive trends, in the same order order of 1–1.5 DU decade$^{-1}$, are retrieved in Mumbai (1.6 ± 0.9 DU decade$^{-1}$), Manila (0.9 ± 0.8 DU decade$^{-1}$), and Kinshasa (1.5 ± 0.8 DU decade$^{-1}$). In other mega-cities, the increase is smaller and insignificant, with the exception of Mexico, where a negative insignificant trend (-1.2 ± 1.2 DU decade$^{-1}$) is found. Ebojie et al. (2016) retrieved a stronger statistically significant ozone increase, of around 2 % year$^{-1}$ in Bangalore (2.3± 1 % year$^{-1}$) and Manila (1.8 ± 1.3 % year$^{-1}$), while the trends from the current study do not exceed 0.7 % year$^{-1}$ (in Bangkok). Additionally, they found a stronger significant decrease in Mexico (-2.0 ± 0.9 % year$^{-1}$ instead of -0.1 ± 0.5 % year$^{-1}$). For the remaining mega-cities, the trends are negative, ranging from -0.2 to -1.5 % year$^{-1}$. Nevertheless, although the

Open Access EGU

**Table 3.** Tropospheric ozone trends in 10 tropical Mega-cities using $CCD_{IUP}$, (in DU decade$^{-1}$ and % year$^{-1}$), $CCD_{DLR}$, and LNM (in % year$^{-1}$) data and tropospheric $NO_2$ trends with their $2\sigma$ uncertainties. The statistically significant trends at the at the 95% confidence ($p < 0.05$) level are shown in bold.

| Site | Trop. $O_3$ trend CCD $DUdec^{-1}$ Current | Trop. $O_3$ trend CCD % $yr^{-1}$ study | Trop. $O_3$ trend $CCD_{DLR}$ $DUdec^{-1}$ Heue et al., 2016 | Trop. $O_3$ trend LNM % $yr^{-1}$ Ebojie et al., 2016 | Trop. $NO_2$ trend $\times10^{15}$ molec.cm$^{-2}$dec$^{-1}$ Schneider et al., 2015 | Trop. $NO_2$ trend $\times10^{15}$ molec.cm$^{-2}$dec$^{-1}$ Hilboll et al., 2013 |
|---|---|---|---|---|---|---|
| Jakarta | -0.2 ± 0.9 | -0.1 ± 0.5 | 0.2 ± 0.8 | -0.2 ± 1.6 | -2.4 ± 1.5 | **-11.9 ± 4.1** |
| Mexico | -0.3 ± 1.9 | -0.1 ± 0.8 | -1.2 ± 1.2 | **-2.0 ± 0.9** | -3.5 ± 2.8 | 5.1 ± 8.2 |
| Manila | **1.1 ± 0.7** | **0.6 ± 0.4** | **0.9 ± 0.8** | **1.8 ± 1.3** | **-3.6 ± 0.7** | **-10.3 ± 2.0** |
| Mumbai | 0.3 ± 1.6 | 0.2 ± 0.8 | **1.6 ± 0.9** | -0.4 ± 0.9 | 0.4 ± 0.8 | **7.0 ± 2.1** |
| Bangkok | **1.4 ± 0.9** | **0.7 ± 0.5** | 0.5 ± 1.0 | – | 2.0 ± 1.6 | – |
| Lagos | 1.1 ± 1.2 | 0.4 ± 0.5 | 0.6 ± 0.9 | **-1.5 ± 1.0** | **2.6 ± 0.5** | **3.3 ± 1.2** |
| Kinshasa | **1.3 ± 0.9** | **0.5 ± 0.4** | **1.5 ± 0.8** | – | 0.4 ± 0.3 | – |
| Bangalore | 0.8 ± 0.9 | 0.4 ± 0.4 | 0.3 ± 0.8 | **2.3 ± 1.0** | **1.9 ± 0.6** | – |
| Lima | -0.5 ± 1.5 | -0.2 ± 0.7 | 0.0 ± 1.1 | – | **3.6 ± 1.1** | **10.6 ± 3.6** |
| Nairobi | 0.7 ± 1.0 | 0.3 ± 0.5 | 0.7 ± 0.8 | – | **1.7 ± 0.4** | – |

trends from the three independent studies do not perfectly agree with each other, they are of the same range (± 2 DU decade$^{-1}$) and within the calculated uncertainties.

The derived tropospheric ozone trends clearly show that the tropospheric ozone increase is not proportional to the population and level of industrialisation of the mega-cities. Schneider et al. (2015) and Hilboll et al. (2013b) studied the $NO_2$ trends over some of the selected mega-cities (see: Table 3). It has been found that $NO_2$ decreases strongly over the largest mega-cities (Jakarta, Mexico, and Manila), revealing that possibly $NO_2$ emissions legislation policies have affected tropospheric ozone concentration. However, there is no direct correlation between tropospheric ozone and $NO_2$ changes shown in Table 3. Emissions of $NO_x$ and VOCs result in ground-level ozone formation. The degree of tropospheric ozone changes strongly depends on the $NO_2$ amount and may decrease (moderate $NO_2$) or enhance ozone (high $NO_2$). Tropospheric ozone changes are also determined by the regional geography and meteorology (meso-scale circulation) that can transport or trap emissions.

## 4 Summary and discussion

The new harmonised data-set of tropical tropospheric ozone columns for the last 20 years between 1996 and 2015, makes it possible to calculate and study long-term tropospheric $O_3$ variability and trends. For this reason, correction offsets have

been calculated for GOME and GOME-2 TTCO using SCIAMACHY as reference (in the middle of the time-series) in order to reduce the instrumental effects in the long-term time series. Nevertheless, the short overlap period between GOME and SCIAMACHY raised limitations in the harmonisation of the GOME dataset. The correction offsets for GOME presented artificial features which are also visible afterwards in the trend. In order to identify the best way to merge the CCD data and also to investigate how the harmonisation approach may affect the observed trends, six different harmonisation scenarios have been tested. The scenario, using no correction for GOME data and the mean bias of GOME-2 with SCIAMACHY as correction offset has been found to show slightly smaller differences comparing to ozonesondes, and, therefore, it is considered to be the reference scenario. After the harmonisation, the data obtained from the different instruments agree better with each other and with the ozonesondes.

Several independent studies have been conducted on tropospheric ozone trends around the world due to its outstanding importance as a greenhouse gas and hazardous pollutant (Cooper et al., 2014; Ziemke et al., 2005; Monks et al., 2015; Oltmans et al., 2013; Lelieveld et al., 2004; Lin et al., 2014; Beig and Singh, 2007; Kulkarni et al., 2010; Heue et al., 2016; Ebojie et al., 2016). However, harmonisation of a multi-instrument dataset is one of the largest sources or uncertainty. Most of the trend studies that use multiple satellite data (e.g. Xu et al. (2011), Loyola et al. (2009), and Heue et al. (2016)) underestimate the uncertainty that harmonisation might introduce, and they calculate their results using only one harmonisation approach. Therefore, in order to quantify the uncertainty due to harmonisation, multi-linear tropospheric ozone trends using all six harmonised datasets have been derived and the maximum deviation between them has been calculated. The trends range between about -4 and 4 DU decade$^{-1}$ and the difference between the trends from the six scenarios has been found to be between 0 and 7 DU, exceeding locally the $2\sigma$ of the trends (0 to 4 DU decade$^{-1}$). We conclude that the statistical regression analysis using the $\beta>2\sigma_\beta$ as criterion to report significant trends in the 95% confidence level is not adequate in order to conclude whether the trends are significant with confidence since the trends uncertainties are larger than the statistical ones.

Using the reference merged dataset, the global tropospheric ozone trend during the period 1996–2015 is found to be almost equal to zero (0.002 % year$^{-1}$) and statistically non significant. This is in agreement with studies of Ziemke et al. (2005) (nearly zero trend) and Ebojie et al. (2016) ($\sim$0.55 DU decade$^{-1}$ or 0.2 % year$^{-1}$) who also found no trend or insignificant trends. This is in contrast with the results of Heue et al. (2016) who found an averaged increase of 0.70 DU decade$^{-1}$. Despite the fact that all the trend results from this study are small (< $\pm$4 DU decade$^{-1}$ or 3 %/ year) and mostly uncertain (66 % are statistically insignificant), there are regions such as over southern Africa, the southern tropical Atlantic, south-east tropical Pacific Ocean, and central Oceania where tropospheric $O_3$ increased significantly by $\sim$3 DU decade$^{-1}$. Additionally, over of central Africa and southern India, tropospheric ozone increased by $\sim$2 DU decade$^{-1}$. Regional positive tropospheric ozone trends of similar magnitude were also observed in other studies (e.g. Lelieveld et al., 2004; Beig and Singh, 2007; Kulkarni et al., 2010; Ebojie et al., 2016; Heue et al., 2016). They might be linked to anthropogenic activities such as emissions in mega cities or biomass burning in combination with changes in meteorology or/and long range transport of the precursor emissions (Hilboll et al., 2013b; Wai et al., 2014; Giglio et al., 2013; Schneider et al., 2015; Cooper et al., 2014; Duncan et al., 2016; Hilboll et al., 2017). On the other hand, tropospheric $O_3$ decreases by $\sim$-3 DU decade$^{-1}$ over the Caribbean sea and parts of North Pacific Ocean, and by less than -2 DU decade$^{-1}$ over some regions of the southern Pacific Ocean. Possible reasons

for this decrease maybe changes in dynamical processes, changes in STE, convection, humidity or precipitation (Morris et al., 2010; Wai et al., 2014; Fontaine et al., 2011; Ebojie et al., 2016; Mieruch et al., 2014; Trenberth et al., 2005; Adler et al., 2003; Chen and Liu, 2016; IPCC, 2007). The biggest limitation interpreting the observed trends over the northern and southern tropical latitudes (>18 $^o$N and S) is the low data sampling at these latitudes. Due to the ITCZ movement, cloudy data during local winters are reduced, making the above cloud ozone column (ACCO) retrieval difficult or violating the invariance of the ACCO per latitude band. Therefore, even though they might appear to be statistically significant, they should be referred to with caution.

Focusing on trends in ten selected mega-cities, a slight tropospheric ozone decrease is observed at the largest cities, such as Jakarta and Mexico ($\sim$-0.3 DU decade$^{-1}$), whereas statistically significant increases ($\sim$ 1 DU decade$^{-1}$) are noticed over Manila, Bangkok, and Kinshasa. It has been shown that tropospheric ozone increase is not linearly related with the size and the industralisation of the selected mega-cities. This is not surprising since tropospheric ozone production from its precursors, $NO_x$ and VOCs, is not linear. For example, very large increase of $NO_x$ or VOCs may result in strong destruction of ozone. It is also broadly recognised that the mechanisms that modulate tropospheric ozone variability are not straightforward according to precursor emission. In addition, meteorological conditions and atmospheric oscillations also play an important role (Ziemke and Chandra, 2003; Solomon et el., 2007; Chandra et al., 2009; Voulgarakis et al., 2010; WMO, 2011; Neu et al., 2014; Monks et al., 2015). Comparing our trend results with Heue et al. (2016) and Ebojie et al. (2016) trend results in these ten mega-cities, we found that they agree slightly better (within the combined uncertainties) with the ones from Heue et al. (2016). The most possible reasons for the mismatch with Ebojie et al. (2016) trends is the fact that their retrieval reaches up to the tropopause including more upper-tropospheric ozone information, and additionally the fact that they investigated a shorter time period, between 2003 and 2012.

The attribution of observed TTCO trends in specific regions to the various processes is not possible without the additional use of chemistry-transport models that can potentially disentangle the different contributions to tropospheric ozone variability (dynamics and chemistry). For example, monthly tagged CTM runs could give insight into tropospheric ozone sources for specific locations (e.g. Coates et al., 2015). With this method, the fate of emitted species is followed and their chemical reaction pathways are tracked. Using labeled CTM mechanisms for $NO_x$ and VOCs emissions, and their degradation products, the ozone burden can be attributed to the relevant emission source (Grewe et al., 2012; Coates et al., 2015).

In the future, the launch of Sentinel 5 precursor satellite (fall 2017) will extend the TTCO record at least for 7 more years (expected S5p lifetime). It is also expected that the extension of the time-series will result in more reliable trend results. The grid box size used in this study was relatively coarse (2.5°×5° degrees), due to the instruments spatial resolution (GOME pixel $\simeq$320 km), and in order to remove the residual noise. The high spatial resolution (7×7 km) of the TROPOMI instrument aboard S5p will improve the trend estimates of tropospheric ozone over mega-cities.

## 5 Data availability

Data used in this publication can be accessed via the IUP website: http://www.iup.uni-bremen.de/UVSAT/datasets or by contacting the corresponding author.

*Acknowledgements.* TEXT

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
