# Peer review of "Harmonisation and trends of 20-years tropical tropospheric ozone data"

_Atmospheric Chemistry and Physics, 2017_

## Referee Comment (RC1) · Anonymous Referee #1 · 7 Nov 2017

General Comments:

This study attempts to harmonize a 20-year satellite record including GOME/ERS-2, SCIAMACHY/Envisat, and GOME-2/MetOp-A using SCIAMACHY as a transfer standard. The authors use 6 different schemes in their attempt to harmonize the data and evaluate the relative success of the various approaches via comparisons with in situ measurements (i.e., sondes) when and where possible. The authors suggest that by using 6 different approaches, they are better able to estimate the uncertainty in apparent trends owed to the harmonization itself.

Like prior studies, the authors find few areas of significant tropospheric ozone trends. Their analysis of trends over tropical mega cities seems to produce results not too different from prior studies. Frankly, I am really not sure what to take away from this

study, and I have several important issues with what is (and is not) presented.

The authors should be aware of a major reprocessing effort of the Southern Hemisphere Additional Ozonesonde (SHADOZ) network data being led by Thompson and Witte. While I realize the papers on their work are just making their way into the literature, their efforts have been ongoing for several years. Since the authors of this study leverage the SHADOZ data, I am surprised that the paper communicates no awareness of the reprocessing effort nor of its potential impact on the results of this study. At the very least, the authors could have contacted SHADOZ PI Thompson to make sure she was aware of this study and had the opportunity to communicate important updates relevant to the authors. While entirely up to the authors, having Thompson as a co-author would have strengthened the credibility of the sonde results presented in this paper.

Finally, while the paper appears to present different approach to tropical tropospheric ozone trend analyses that recent studies, I do not find the results particularly compelling or worthy of publication in this form a this time.

Recommendation:

I recommend this manuscript be declined for publication in ACP at this time pending major revisions.

Detailed Comments:

Page 1 Line 3: What does "good agreement" mean? Quantify. Line 14: "Additionally, over central..." Awkward sentence. Line 19: "...reasons for these decreases are..."

Page 2 Line 2: delete "both"

Page 3 Line 10: "3.8% decade-1 (0.16 ppbv year-1)" What is the difference between these numbers? Unclear. You mention "surface and ozonesonde observations," so which is it?

Page 4 Line 4: The reference to Leventidou et al., 2016 may be a recent one for the CCD method, but I think you should be referencing the original paper for this approach, which I believe goes back to Ziemke... Line 10ff: This would be a good place to remind the reader of the specific application of the CCD approach you're using. To what altitude is tropospheric column ozone being computed? Does it vary scene-to-scene? On what fraction of pixels can it be applied? Line 14: "overpass time" – is this not a critical element influencing tropospheric ozone, especially in regions near megacities? Line 25: "...whole timespan of the operation of the European satellites..." Line 26: "...since it is the only..."

Page 5 Line 2: "Possible reasons for the biases are..." The paper is filled with these statements. It would be good to know if this is the problem. Could you test your hypothesis by applying the same cloud algorithm to both retrievals? I realize that requires working with the instrument teams, but even a limited test application could prove useful. The differences that appear between the two panels of Figure 1 are striking. To me, this subject is more interesting than the one that is the current main focus of the paper. Figure 1. It occurs to me in looking at the upper panels that perhaps it would be worthwhile to separate the lower plots into "over land" and "over water" components. Visually, It would also be helpful to the reader if they plots were rotated 90 deg. so that the latitudes ran up and down the page as they do in the top panel. Line 8 – 9: "This behavior may be explained by the short time of common operation..." Another undemonstrated hypothesis. How could you test this assumption? And if it's true, is not your transfer standard idea (i.e., reference to SCIAMACHY) compromised?

Page 6 Line 3: "...ozonesonde data, it seems reasonable..." Figure 2. What is important about this plot is how little of the area is actually statistically significant. Perhaps you should reverse and only mark with "x" those cells that ARE statistically significant. Also, remind the reader, what fraction of the cells are statistically significant? This result is key in your argument that you can use a constant offset to "correct" the GOME 2 data, but you do not seem to make much of a point of that in the text. Line 12: "..but

only a subset of them." Which subset? Which years? Sect. 2.3: This seems to be an important part of the paper, but frankly, I do not find it well motivate. While 6 scenarios? Have you exhausted all possibilities. Does the reader need to know the details or simply your recommendation for the best approach to harmonize the data – with a discussion of the other approaches you tried and how they inform your estimate of the component of the calculated trend uncertainties that arise from the harmonization process itself?

Page 7 Table 1. Not sure what to do with this table. Why are the stations in the order in which they appear? What is the table communicating? Here's where my earlier comment about SHADOZ reprocessing becomes relevant: are you using the reprocessed sonde data? How would these results change if you did? You are integrating the sondes to 200 hPa. How does that compare to the altitude used for the CCD approach?

Page 8 Line 7: "The same occurs for the..." delete the comma. Line 9: "...the scenarios that can be confidently rejected according to this comparison are..." I'm not sure I have confidence that any scenario can be rejected until I know more about the sonde data you used and the altitudes used for the satellite tropospheric column amount.

Page 9 Line 1: "where $\alpha$ is the offset..." Line 12: "...to persist into the next month." Line 16: define "AR(1)" Lines 25ff: "Nevertheless, all scenarios shown in Fig. 3 agree that there is a positive trend..." My quick read of Figure 3 is that very little of the map shows statistically significant trends. As to the fact that one appears to exist "over the southern tropical Atlantic Ocean" and a couple of other sites cited by the author, I am not sure what to make of it. The authors provide no explanation for such trends or why they might exist. I would find this more compelling if the authors could simplify the presentation, show the best correction scheme, show the best estimation of uncertainty (including that resulting from the harmonization scheme), and then spent some time in the text discussion what the resulting trend data showed and why. In its current form, I find the presentation more confusing than compelling.

Page 11 Line 8: "...following very well a Gaussian distribution." You don't show that in the paper. But I'm not sure what to make of it, either. Are you saying that there is no signal anywhere on the map? What do you mean by "the noise is random?" What is the noise? Line 10ff: "This result is in agreement with Ziemke et al. (2005) and Ebojie et al. (2016)..." What periods did they examine? What data did they use? Are there any implications from the fact that it does not appear to have changed from their analyses to your analysis? What new have we learned from your analysis?

Page 12 Line 10: "Figure 5 summarizes the tropical tropospheric ozone trends..." Line 25: "...may still be an artifact of the data-set." Is there a way to know? Line 31ff: Just to be clear, you selected your regions based on where you found statistical significance? That led to larger regions that then had statistically significan trends? Your Table 2 shows some impressive trend results. You then follow that with a list of possible factors that led to the trends (anthropogenic NOx, population, energy consumption, biomass burning, changes in meteorology, dynamical oscillations, stratospheric intrusions) and you cite some prior works that have made these suggestions, but you provide no evidence within this paper for the proximal cause (or causes) in each of the regions you list in Table 2, nor is there really any justification for the selection of the boundaries of those regions other than they produce significant trends. If, as the title of this section suggests, mega-cities are responsible, it seems the regions might have been more narrowly defined. It would have been nicer to select regions based on a hypothesis and then identify the existence of significant trends (or not) to accept or reject that hypothesis.

Page 16 Line 10: "Despite the fact that might appear to be..." What might appear? Lines 22ff: "...cloudiness and humidity which contribute to photochemical O3 loss..." I think the missing factor identified in the Morris et al., 2010 paper was significant lightning production, which they hypothesized led to NOx production and O3 loss in the absence of sunlight. The presentation here is a bit oversimplified. Deep convection alone can loft relatively low O3 concentrations from near the surface (especially over

the sea) to the upper troposphere. Those decreases are not "loss" but reductions resulting from transport.

Page 17 Line 2: "winter" – what does "winter" mean in the tropics? Perhaps it's better to identify seasons by months rather than such ambiguous names. Line 3: "as NO2 over North America and Europe may have affected the O3 trends. . ." moved the comma. Line 8ff: "Possible reasons. . ." There's a whole list of possibilities here with no conclusions or evidence to support any one (or combination) of them. Line 10: ". . .water vapor in the troposphere accounts for one of the most important. . ." Lines 10 – 12: "An increase in vertical convective patterns over the tropical oceans may result in lower ozone mixing ratios in the upper troposphere. . ." True if lofting low ozone from the surface. If lightning is present in the convection, however, you might see enhancements. Thus, the influence is unclear.

Page 18 Lines 23ff: I would replace all of this text with a table. Non need to write it all out.

Page 19 Table 3. Like previous tables, what is the logic of the order of cities in this table? How do the periods of study for Heue, Ebojie, Schneider, HIlboll, and this work compare? What impacts do differences in study periods have on interpretation of the results? The data in this table appear to have been compiled using 2.5 X 5 deg. boxes. That's roughly an area 250 km X 500 km in the tropics. Can you actually see signatures from megacities spread out over such a large area? For a control, should you also compute trends around cities that have not grown (perhaps ones that have shrunk) or that have reduced emissions just to see if they behave any differently than the ones you list here? Line 3: "The derived tropospheric trends clearly show that the tropospheric ozone increase is not proportional to. . ." If you're going to make this claim, I think you need to show the population data, perhaps in Table 3, and the proxy you're using for industrial activity as well. Lines 8 – 9: "The degree of tropospheric ozone change strongly depends on the NO2 amount. . ." As the second half of this sentence correctly relates, it depend on the relative NO2 and VOC concentrations. I think I would

get rid of the word "strongly" in this sentence.

Page 20 Line 21: "...since the uncertainties in the trends are larger..." Line 24: Cite the uncertainties associated with the trends published in Ebojie et al. Line 31 – 32: "They might be linked to..." You list a whole bunch of possibilities. Has anyone shown the specific relevant link for your study? If not, how can you test these hypothetical influences?

Page 21: Line 4: "...tropical latitudes (> 18 0N and S)..." What range? Line 10: "It has been shown that tropospheric ozone increase is not linearly related..." I'm not sure this study has shown that result conclusively or persuasively. Line 18: "...the fact that their retrieval reaches up to the tropopause..." That seems like an important factor. How different are the retrievals? What impact do the differences have on your results/homogenization scheme? Line 28: "...(expected lifetime of the Sentinel 5 precursor satellite)." No need to introduce an abbreviation in the last paragraph.
* * *

---

## Referee Comment (RC2) · Anonymous Referee #2 · 14 Dec 2017

The authors present in the paper a detailed trend analysis of tropospheric ozone over the tropics, using a long term homogenized data set based on satellite measurements using the Convective Clouds Differential method. This method and its application on individual satellite sensors has already been presented in various studies including a publication from the same group in AMT (Leventidou et al. 2016). In this paper they homogenize the data from three sensors and examine the variability and the trends over regions and mega cities within the tropics. The paper is well written and structured but there are many significant issues that should be considered before being accepted for publication in ACP.

The content of the paper has many similarities with the paper by Heue et al. 2016 in

[Figure]

AMT. Although it is clear that the Heue et al., paper uses a different version of total ozone data it is not clear from the current paper what are the differences between these two data sets concerning the application of the CCD method and the resulting tropospheric ozone estimates. The authors should elaborate more here.

Section 2.2 The authors attribute most of the differences between the TTCO mostly to the different cloud algorithms involved. Why they exclude eventual biases between the sensors also in the initial total columns? Is there any explanation for the different behavior if GOME-SCIA differences over 10oN shown in Figure 1? The authors should also provide an explanation for the GOME-2/SCIA drift. Does this originate from a potential drift in the total columns?

Section 2.3. The discussion of six scenarios in the paper is confusing, since they don't differ substantially concerning the outcome. I think the authors should just describe here the chosen approach of harmonization.

Section 3.3.1. The authors present regional trends in this sections. The choice of the regions to my understanding is based only on the significance of the trends and in a sense this looks like a random choice. Do these regions have some special characteristics that have to do either with prevailing dynamic features or emission sources? In general the discussion here should be improved.

Section 3.3.2. The authors attribute the positive trends to South Africa and South America to biomass burning. Is there any indication from another source that there is increased biomass burning over the years that can cause such a trend?

Section 3.3.3. The authors show trends over mega-cities in the tropics. The authors should provide a comment why they think a grid-point of 2x5degrees can represent the variability of tropospheric ozone caused by a mega city. The discussion against NO2 trends as shown in the paper is also not conclusive. Are there studies (modelling or in-situ ones) to support their findings? The authors also compare their results with Heue et al and although the approach is pretty similar they are differences. They should

elaborate more here to explain this.

---

## Author Comment (AC1) · 2 Feb 2018

Answers on Anonymous Referee #1

General Comments:

This study attempts to harmonize a 20-year satellite record including GOME/ERS-2, SCIAMACHY/Envisat, and GOME-2/MetOp-A using SCIAMACHY as a transfer standard. The authors use 6 different schemes in their attempt to harmonize the data and evaluate the relative success of the various approaches via comparisons with in situ measurements (i.e., sondes) when and where possible. The authors suggest that by using 6 different approaches, they are better able to estimate the uncertainty in apparent trends owed to the harmonization itself. Like prior studies, the authors find few areas of significant tropospheric ozone trends. Their analysis of trends over tropical mega cities seems to produce results not too different from prior studies. Frankly, I am really not sure what to take away from this study, and I have several important issues with what is (and is not) presented.

The authors should be aware of a major reprocessing effort of the Southern Hemisphere Additional Ozonesonde (SHADOZ) network data being led by Thompson and Witte. While I realize the papers on their work are just making their way into the literature, their efforts have been ongoing for several years. Since the authors of this study leverage the SHADOZ data, I am surprised that the paper communicates no awareness of the reprocessing effort nor of its potential impact on the results of this study.

At the very least, the authors could have contacted SHADOZ PI Thompson to make sure she was aware of this study and had the opportunity to communicate important updates relevant to the authors. While entirely up to the authors, having Thompson as a co-author would have strengthened the credibility of the sonde results presented in this paper.

Finally, while the paper appears to present different approach to tropical tropospheric ozone trend analyses that recent studies, I do not find the results particularly compelling or worthy of publication in this form a this time.

Recommendation:

I recommend this manuscript be declined for publication in ACP at this time pending major revisions.

*Our answer:*
*Many thanks for the very helpful comments. As proposed by the reviewer, we have contacted Drs. Anne Thompson and Bryan Johnson who are responsible for reprocessing the ozonesondes data from SHADOZ network. Both accepted our invitation to become co-authors of the paper and supported us in revising the paper. We now used reprocessed SHADOZ data (avaliable for American Samoa and Paramaribo). The recent reprocessing mainly focused on improving old sonde data at the stratospheric ozone peak and above. It had very minor impact on our findings. The stations of Hilo and Fiji have been removed from the comparisons. Fiji is affected by air masses originating from the*

*mid-latitudes and the upper troposphere and Hilo is strongly affected by volcanic outgassing, resulting in negligible ozone concentrations in the boundary layer.*

*Additionally, Klaus Peter Heue is included as co-author in the paper for granting us access to his tropospheric ozone data which we used for some comparisons.*

*Since Reviewer #2 suggested to shorten the discussion on the various merging approaches we moved some material into a supplement and discuss mainly the lessons learned from looking at trends from differently merged datasets in Section 2. The estimation of the mean tropical trends is moved to Subsection 4.1 and is now limited between 15°S and 15°N since the tropical borders are strongly influenced by air masses being transported from the mid-latitudes and stratospheric intrusions (Thompson et al., 2017).*

*Section 3.3.3. about trends in megacities has been removed from the revised version of our paper as also suggested by reviewer #2.*

*At various places we have expanded on the comparisons to the Heue et al. results. Although similar instruments have been used, the results from this study and Heue et al. are different and are discussed in more detail (see detailed comments).*

*Our main findings can be summarized as follows:*
*Tropical tropospheric ozone trends critically depend on the merging/harmonisation approach. This was investigated by investigating six different merging scenarios. The trend of tropical tropospheric ozone is estimated using a multiple linear regression model and for all six scenarios the sensitivity of the derived trends to the harmonisation approach is investigated. Such an approach has not been reported before and may explain why tropospheric ozone trends from different studies do not agree (see e.g. TOAR report). The main conclusion is that the (statistical) trend uncertainties from one scenario may be smaller than the variation of trends from the different merging approaches, which means that the trend uncertainties are in reality larger. At the end we selected the preferred merging scenario by comparing these six merged datasets with ozonesonde data (some of them reprocessed now) from the SHADOZ network.*

Detailed Comments:

**Page 1**
--Line 3: What does "good agreement" mean? Quantify.

*We consider the bias between the CCD retrievals and the integrated $O_3$ profiles from ozone sondes good since they are less than 6 DU which is about the 1sigma uncertainty of the mean station bias (RMS in Table 2 of Leventidou et al., 2016).*

*We changed in the main text (page 4, line 30) as follows: "The biases between them have been found to be within 6 DU which is mostly within the uncertainties of the mean biases of 6 DU (1 sigma). One large source of uncertainties in these comparisons are the*

*low sampling of the sondes (less than five launches in a month typically) and the fact that CCD ozone is only derived as monthly means covering rather large areas (grid boxes). In the abstract we only mention the average bias between CCD and sondes".*

Line 14: "Additionally, over central ..." Awkward sentence.
*The sentence has been changed to "... and by ~2DU/ decade over central ..."*

Line 19: "… reasons for these decreases are..."
*The sentence has been removed.*

**Page 2**
--Line 2: delete "both"
**Deleted**

**Page 3**
--Line 10: "3.8% decade-1 (0.16 ppbv year-1)" What is the difference between these numbers? Unclear. You mention "surface and ozonesonde observations," so which is it?

*The trend in ppbv year$^{-1}$ represents the change of surface ozone in volume mixing ratio per year. The sentence (page 3, line 5) has been changed to: "Oltmans et al. (2013) observed an increase of 3.8 % decade$^{-1}$ (0.16 ppbv year$^{-1}$) in surface ozone in Mauna Loa, Hawaii (19.5◦N) in the North Pacific since 1974 and a smaller insignificant trend in the order of 0.7 % decade$^{-1}$ (0.01 ppbv year$^{-1}$) in American Samoa (14.5$^{◦}$S) after 1976. "*

**Page 4**
--Line 4: The reference to Leventidou et al., 2016 may be a recent one for the CCD method, but I think you should be referencing the original paper for this approach, which I believe goes back to Ziemke ..

*As the referee mentions, the CCD method was developed by Ziemke et al., 1998 and further improved by Valks et al., 2003. The citation of Ziemke et al. 1998 has been added in the introduction, along with the most significant contributors on tropospheric ozone retrievals from remote sensing in the past.*

--Line 10ff: This would be a good place to remind the reader of the specific application of the CCD approach you're using. To what altitude is tropospheric column ozone being computed? Does it vary scene-to-scene? On what fraction of pixels can it be applied?

*The CCD method is described in detail in Leventidou et al., 2016. All tropospheric O3 columns are calculated up to 200 hPa. The main reason is that most clouds do not reach the tropopause.*

--Line 14: "overpass time" – is this not a critical element influencing tropospheric ozone, especially in regions near megacities?

*For tropospheric trace gases that show diurnal variations, overpass time is important. All satellite data used here are in the morning hours differing at most one hour (9:30 to 10:30). This is believed to have little impact on the results.*

**The following text has been added (page 9, line8): "As seen in Table 1, the mean bias between the six harmonised TTCO datasets and the ozone sondes range between -1.1 and 0.9 DU which is well within the retrieval uncertainty showing that for most scenarios the spatio-temporal offsets with respect to ozonesondes are minimised."**

--Line 25: " … whole timespan of the operation of the European satellites…"
*changed*

--Line 26: "...since it is the only..."
*changed*

**Page 5**
--Line 2: "Possible reasons for the biases are..." The paper is filled with these statements. It would be good to know if this is the problem. Could you test your hypothesis by applying the same cloud algorithm to both retrievals? I realize that requires working with the instrument teams, but even a limited test application could prove useful. The differences that appear between the two panels of Figure 1 are striking. To me, this subject is more interesting than the one that is the current main focus of the paper.

*It would be desirable to have the same cloud algorithm for all instruments. However, any bias from the cloud algorithm has been removed by the harmonisation process. We believe that the different spatial resolutions of the instruments is more important (GOME :320 x 40 km$^2$, SCIAMACHY: 60 x 30 km$^2$, and GOME-2: 80 x 40 km$^2$).*

--Figure 1. It occurs to me in looking at the upper panels that perhaps it would be worthwhile to separate the lower plots into "over land" and "over water" components. Visually, It would also be helpful to the reader if the plots were rotated 90 deg. so that the latitudes ran up and down the page as they do in the top panel.

*In order to keep it simple we leave the figure as is. The land-sea contrast can be clearly seen in the 2D plots. The error bars in Fig. 1 (line graphs) mainly reflect the longitudinal variation possibly due to land-sea contrast. The line graphs have been changed so that the y-axis is the latitude.*

--Line 8 – 9: "This behavior may be explained by the short time of common operation..." Another undemonstrated hypothesis. How could you test this assumption? And if it's true, is not your transfer standard idea (i.e., reference to SCIAMACHY) compromised?

*The larger variation in the bias with latitude in GOME data is most likely due to the short overlap period (10 months, from August 2002 to June 2003 (when GOME lost its global coverage)). For GOME-2 the overlap with SCIAMACHY was more than 5 years, making the latitude dependence smoother.*

**Page 6**
--Line 3: "...ozonesonde data, it seems reasonable..."
*changed*

--Figure 2. What is important about this plot is how little of the area is actually statistically significant. Perhaps you should reverse and only mark with "x" those cells that ARE statistically significant.
Also, remind the reader, what fraction of the cells are statistically significant? This result is key in your argument that you can use a constant offset to "correct" the GOME 2 data, but you do not seem to make much of a point of that in the text.

*Figure 2 has been changed, showing with x the statistically significant grid boxes. It is clear from the figure that the vast majority of the grid points are statistically insignificant and there is no need to specify.*

*Line 14, page 9 on the following modified sentence:"Scenario 6 can also be rejected due to the fact that the drift in GOME-2 correction offset at 81% of the grid-boxes is statistically non significant. "*

*We have added a line plot to Fig. 2 to show that the drift is not significant. The reviewer is correct that this is the main reason that a drift correction is not needed as already mentioned in the text.*

--Line 12: "..but only a subset of them." Which subset? Which years?

.....of the ITCZ (no cloudy data available in the western Pacific)

*The sentence has been removed since the explanation is given the sentence before.*

--Sect. 2.3: This seems to be an important part of the paper, but frankly, I do not find it well motivated. Why 6 scenarios? Have you exhausted all possibilities. Does the reader need to know the details or simply your recommendation for the best approach to harmonize the data – with a discussion of the other approaches you tried and how they inform your estimate of the component of the calculated trend uncertainties that arise from the harmonization process itself?

*We think that this is one of the most important results of this paper. We show here that the hamonisation procedure (merging) is one of the largest error sources of the trends. In the six scenarios we checked different reasonable assumptions on how to handle the differences between the individual instruments. The trends derived from the various merged dataset show larger differences than the statistical uncertainty from the trend regression applied to one of them. This is usually neglected in other studies.*

*To make this section a bit shorter as also suggested by Reviewer #2, we moved*

*this figure showing the maximum trend difference among all six merging scenarios to the supplementary material. Fig. S2 shows that the mean differences in trends from all pairs of merged datasets is about 2 DU/decade, exceeding in most cases the uncertainty from the single data regression.*

**Page 7**
--Table 1. Not sure what to do with this table. Why are the stations in the order in which they appear? What is the table communicating? Here's where my earlier comment about SHADOZ reprocessing becomes relevant: are you using the reprocessed sonde data? How would these results change if you did? You are integrating the sondes to 200 hPa. How does that compare to the altitude used for the CCD approach?

*Table 1 showed comparisons between integrated ozone columns up to 200hPa from 9 tropical ozonesonde stations from the SHADOZ network (version V05) with 6 different possible merging scenarios of Tropical tropospheric ozone columns from GOME, SCIAMACHY, and GOME-2. The tropospheric ozone columns retrieved with our CCD algorithm are adjusted to 200 hPa using climatological values (Leventidou et al., 2016).*

*The change in the differences of tropospheric ozone columns  (up to 200 hPa) between collocated  CCD_results and SHADOZ for the stations of Paramaribo, Am. Samoa, Hilo, and Fiji  due to changes from SHADOZ V05 to  V05.1R .*

| Station | Scenario 1 | Scenario 2 | Scenario 3 | Scenario 4 | Scenario 5 | Scenario 6 |
|---------|------------|------------|------------|------------|------------|------------|
| Am. Samoa | 0.6 | 1.0 | 0.6 | 0.0 | 0.7 | 4.3 |
| Paramaribo | 1.8 | 0.7 | 1.7 | 2.7 | 1.8 | -1.1 |

*Following the comments of the reviewer the comparison has been updated including the newest version of SHADOZ data (for two stations). The stations of Fiji and Hilo have been removed from the paper as discussed earlier. The following text has been added:" Fiji (18.1S, 178.4E)) station is not included in the comparison because it is highly influenced by air coming in from mid-latitudes and the upper troposphere (Thompson et al., 2017). Hilo (19.4N, 155.4W) is influenced by volcanic out-gassing with high SO2 emissions, resulting in negligible ozone concentrations at the boundary layer. Therefore, this station is also not included."*

*The order of the stations has changed to alphabetical, and the title and the table has changed as follows:*

*"Mean differences (in DU) between merged TTCO data, retrieved with the CCD method using six possible harmonisation scenarios, with integrated ozone columns up to 200 hPa from nine SHADOZ stations. The stations marked with asterisk present data from the newest reprocessed (V05.1_R) version (Thompson et al., 2007; Witte et al., 2017). The regions where the merged scenarios have the smallest biases with the ozonesondes are marked with bold. Scenario 1 has the smallest mean bias for all the stations."*

| Station | Scenario 1 | Scenario 2 | Scenario 3 | Scenario 4 | Scenario 5 | Scenario 6 |
|---------|------------|------------|------------|------------|------------|------------|

| | | | | | | |
|---|---|---|---|---|---|---|
| Am. Samoa[*] | -0.89 | -0.92 | -1.99 | -0.61 | -0.93 | 4.59 |
| Ascension | 0.03 | -0.14 | -0.77 | -0.42 | -0.6 | 0.03 |
| Java | -0.11 | -0.12 | -1.12 | -0.54 | -0.55 | -0.11 |
| Kuala Lumpur | -1.81 | -2.12 | -2.12 | -2.14 | -2.48 | -1.78 |
| Nairobi | 1.81 | 1.10 | 1.80 | 1.48 | 0.74 | 1.84 |
| Natal | 0.56 | 0.63 | -0.21 | 0.22 | 0.28 | 0.57 |
| Paramaribo[*] | -2.98 | -2.95 | -3.02 | -4.11 | -4,34 | -0.11 |
| Mean bias for all stations | -0.48 | -0.64 | -1.06 | 0.87 | -1.13 | 0.72 |

*The results with the updated ozonesonde data do not differ significantly from our earlier results. Nevertheless, we present our results now using the updated V05_R1 ozonesonde data for the available stations.*

**Page 8**
--Line 7: "The same occurs for the..." delete the comma.
*changed*

--Line 9: "...the scenarios that can be confidently rejected according to this comparison are..." I'm not sure I have confidence that any scenario can be rejected until I know more about the sonde data you used and the altitudes used for the satellite tropospheric column amount.

*The updated ozonesonde data to not change the conclusions. The tropospheric ozone columns from the sonde data were calculated exactly as the satellite data up to 200 hPa.*

*The text has been changed as follows(page 9, line 14):*
*" …. Although the comparison between the TTCO from the individual harmonised scenarios and the ozonesonde data does not favor clearly any harmonisation scenario, the scenarios that can be confidently rejected are scenarios 3, 4 and 5 where GOME data are corrected with respect to SCIAMACHY since the overlap period between GOME and SCIAMACHY is very short (10 months, 8/2002-6/2003). Scenario 6 can also be rejected due to the fact that the drift in GOME-2 correction offset at 81% of the grid-boxes is statistically non significant. Lack of significant drifts in the comparison between GOME-2 and SCIAMACHY over the overlapping period shows that the data records are quite stable. Finally, scenario 1 (no drift corrections and bias correction for GOME-2) has the smallest mean bias with the ozone sondes (-0.4 DU). For these reasons, scenario 1 has been selected to be the preferred harmonisation scenario for merging the TTCO datasets."*

**Page 9**
--Line 1: "where $\alpha$ is the offset ..."
*Changed*

--Line 12: "...to persist into the next month."

*changed*

--Line 16: define "AR(1)"
*The sentence has been changed as: "Therefore, the first order autocorrelation of the noise (AR[1]) is included in the model, as explained by Weatherhead et al. (1998)."*

--Lines 25ff: "Nevertheless, all scenarios shown in Fig. 3 agree that there is a positive trend..." My quick read of Figure 3 is that very little of the map shows statistically significant trends. As to the fact that one appears to exist "over the southern tropical Atlantic Ocean" and a couple of other sites cited by the author, I am not sure what to make of it. The authors provide no explanation for such trends or why they might exist. I would find this more compelling if the authors could simplify the presentation, show the best correction scheme, show the best estimation of uncertainty (including that resulting from the harmonization scheme), and then spent some time in the text discussion what the resulting trend data showed and why. In its current form, I find the presentation more confusing than compelling.

*Figure 3 e) shows the regions where the statistically significant  trends calculated using the preferred merging scenario exceed the maximum difference of the trends among all six merging scenarios (now Fig S2) and can be reported with the highest confidence.*

**Page 11**
--Line 8: "… following very well a Gaussian distribution." You don't show that in the paper. But I'm not sure what to make of it, either. Are you saying that there is no signal anywhere on the map? What do you mean by "the noise is random?" What is the noise?

*This sentence has been removed as it is out of context here.*

--Line 10ff: "This result is in agreement with Ziemke et al. (2005) and Ebojie et al. (2016)… " What periods did they examine? What data did they use? Are there any implications from the fact that it does not appear to have changed from their analyses to your analysis? What new have we learned from your analysis?

*The  refereed sentence (now page 11, line 30) has been modified as follows:*
*"The mean tropospheric ozone trend is in agreement with Ziemke et al. (2005) (using solar backscatter ultraviolet (SBUV) and Total Ozone Mapping Spectrometer (TOMS) version data from 1979 to 2003) and Ebojie et al. (2016) (using SCIAMACHY limb-nadir-matching (LNM) observations during the period 2003–2011) who also indicated insignificant and near zero global trends in the tropics, although their analysis was based on different datasets and covered shorter time periods. "*

**Page 12**
--Line 10: "Figure 5 summarizes the tropical tropospheric ozone trends ..."
*changed*

--Line 25: "…may still be an artifact of the data-set." Is there a way to know?

*The sentence (now page 11, line 18)  has changed as follows: " The negative trends appearing in a region at the northern latitudes (Caribbean sea and northern Pacific) may be an artifact of the data-set (low sampling of data,  54 out of 240 months of data)."*

--Line 31ff: Just to be clear, you selected your regions based on where you found statistical significance? That led to larger regions that then had statistically significant trends?

*Yes, we selected the regions in order to have large number of grid points with significant trends for highlighting.*

--Your Table 2 shows some impressive trend results. You then follow that with a list of possible factors that led to the trends (anthropogenic NOx, population, energy consumption, biomass burning, changes in meteorology, dynamical oscillations, stratospheric intrusions) and you cite some prior works that have made these suggestions, but you provide no evidence within this paper for the proximal cause (or causes) in each of the regions you list in Table 2, nor is there really any justification for the selection of the boundaries of those regions other than they produce significant trends. If, as the title of this section suggests, mega-cities are responsible, it seems the regions might have been more narrowly defined. It would have been nicer to select regions based on a hypothesis and then identify the existence of significant trends (or not) to accept or reject that hypothesis.

*The conversation about the possible reasons for the noticed trends has been moved in the conclusions. This section now summarises the areas where we observe statistically significant TTCO trends and compares these results with other studies.*

**Page 16**
--Line 10: "Despite the fact that might appear to be..." What might appear?

*The sentence has changed as follows (page 14, line14): "However,  the observed trends over the northern and southern tropical latitudes ($18^o$ –$20^o$ in SH and NH) should be generally interpreted with caution because they are influenced by low sampling of data due to the movement of the ITCZ, which reduces the cloudy data during local winters and makes the above cloud ozone column (ACCO) retrieval difficult, violating in some cases the invariance of the ACCO per latitude band."*

--Lines 22ff: "...cloudiness and humidity which contribute to photochemical O3 loss..." I think the missing factor identified in the Morris et al., 2010 paper was significant lightning production, which they hypothesized led to NOx production and O3 loss in the absence of sunlight. The presentation here is a bit oversimplified. Deep convection alone can loft relatively low O3 concentrations from near the surface (especially over the sea) to the upper troposphere. Those decreases are not "loss" but reductions resulting from transport.

*We removed most of the discussions on possible causes as we can only speculate on it. In the Summary (page 18, line 34) we briefly mention that we cannot attribute the observed changes in tropospheric ozone as numerous factors may contribute the trends (production, loss, transport). Only with the help of modelling data one can disentangle the various factors.*

**Page 17**
--Line 2: "winter" – what does "winter" mean in the tropics? Perhaps it's better to identify seasons by months rather than such ambiguous names.
*Changed*

--Line 3: "as NO2 over North America and Europe may have affected the O3 trends..." moved the comma.
*Changed*

--Line 8ff: "Possible reasons… " There's a whole list of possibilities here with no conclusions or evidence to support any one (or combination) of them.

*The speculations about possible reasons that are responsible for the observed trends has been removed from the text. Instead a paragraph has been added in the summary where the complexity of trends' interpretation is discussed.*

--Line 10: "...water vapor in the troposphere accounts for one of the most important..."
*Changed*

--Lines 10 – 12: "An increase in vertical convective patterns over the tropical oceans may result in lower ozone mixing ratios in the upper troposphere…" True if lofting low ozone from the surface. If lightning is present in the convection, however, you might see enhancements.
Thus, the influence is unclear.

*See our earlier reply above (has been removed)*

**Page 18**
--Lines 23ff: I would replace all of this text with a table. No need to write it all out.

*Section 3.3.3. and the discussion about trends in mega cities has been removed from the paper as also suggested by Reviewer #2.*

**Page 19**
--Table 3. Like previous tables, what is the logic of the order of cities in this table? How do the periods of study for Heue, Ebojie, Schneider, HIlboll, and this work compare? What impacts do differences in study periods have on interpretation of the results? The data in this table appear to have been compiled using 2.5 X 5 deg boxes. That's roughly an area 250 km X 500 km in the tropics. Can you actually see signatures from megacities spread out over such a large area? For a control, should you also compute trends around cities that have not

grown (perhaps ones that have shrunk) or that have reduced emissions just to see if they behave any differently than the ones you list here?
*See previous reply*

--Line 3: "The derived tropospheric trends clearly show that tropospheric ozone increase is not proportional to ..." If you're going to make this claim, I think you need to show the population data, perhaps in Table 3, and the proxy you're using for industrial activity as well.
*See previous reply*

 --Lines 8 – 9: "The degree of tropospheric ozone change strongly depends on the NO2 amount…" As the second half of this sentence correctly relates, it depends on the relative NO2 and VOC concentrations. I think I would get rid of the word "strongly" in this sentence.
*See previous reply*

**Page 20**
--Line 21: "… since the uncertainties in the trends are larger..."
*See previous reply*

--Line 24: Cite the uncertainties associated with the trends published in Ebojie et al.
*See previous reply*

--Line 31 – 32: "They might be linked to..." You list a whole bunch of possibilities. Has anyone shown the specific relevant link for your study? If not, how can you test these hypothetical influences?

*The  speculations about possible reasons that are responsible for the observed trends has been removed from the text. Instead a paragraph has been added in the summary where the complexity of the interpretation of the trends is discussed. It is out of the scope of this paper to attribute the trends to specific processes.*

**Page 21**
--Line 4: " ...tropical latitudes (> 18 0N and S)..." What range?
*The sentence has changed as follows (page 18, line13): "The most important limitation in interpreting the observed trends over the northern and southern tropical latitudes (18o– 20o in SH and NH) is the low data sampling at these latitudes. "*

--Line 10: "It has been shown that tropospheric ozone increase is not linearly related..." I'm not sure this study has shown that result conclusively or persuasively.

*This text has been removed (mega cities)*

--Line 18: "...the fact that their retrieval reaches up to the tropopause ..." That seems like an important factor. How different are the retrievals? What impact do the differences have on your results/homogenization scheme?

*It is true that we cannot directly compare the trend results but, as mentioned earlier, we use them as indications of the range of the estimated trends.*

--Line 28: "...(expected lifetime of the Sentinel 5 precursor satellite)." No need to introduce an abbreviation in the last paragraph.

*changed*

---

## Author Comment (AC2) · 2 Feb 2018

The authors present in the paper a detailed trend analysis of tropospheric ozone over the tropics, using a long term homogenized data set based on satellite measurements using the Convective Clouds Differential method. This method and its application on individual satellite sensors has already been presented in various studies including a publication from the same group in AMT (Leventidou et al. 2016). In this paper they homogenize the data from three sensors and examine the variability and the trends over regions and mega cities within the tropics. The paper is well written and structured but there are many significant issues that should be considered before being accepted for publication in ACP.

The content of the paper has many similarities with the paper by Heue et al. 2016 in AMT. Although it is clear that the Heue et al., paper uses a different version of total ozone data it is not clear from the current paper what are the differences between these two data sets concerning the application of the CCD method and the resulting tropospheric ozone estimates. The authors should elaborate more here.

*Our reply:*

**Many thanks for the comments.**

**The discussion on the various merging approaches is shortened and a part of it is moved into a supplement. Now we discuss mainly the lessons learned from looking at trends from differently merged datasets in Subsection 2.4. The estimation of the mean tropical trends is now limited between 15°S and 15°N since the topospheric ozone retrievals are questionable (also for the case of Heue et al. (2016)) due to the fact that the tropical borders are strongly influenced by air masses being transported from the mid-latitudes and stratospheric intrusions (Thompson et al., 2017).**

**At various places we have expanded on the comparisons to the Heue et al. (2016) results. Although similar instruments have been used, the results from this study and Heue et al. (2016) are different and are discussed in more detail. In page 4, line 6 we added: "The main differences between our CCD algorithm and the one developed by Heue et al. (2016) originate from the corrections that we have applied in the above cloud column calculation of GOME and GOME-2 data and handling of the outlier data (Leventidou et al., 2016)."**

**Our study shows that despite the fact that the same instruments are used, the trends differ. These differences can be attributed to the different harmonization/merging approaches applied in addition to the different ozone and cloud retrievals used. This paper clearly shows that the merging approach is rather a large source of uncertainty in determining tropospheric ozone trends. This is in our opinion is demonstrated for the first time in this paper.**

Detailed comments:

Section 2.2
The authors attribute most of the differences between the TTCO mostly to the different cloud algorithms involved. Why they exclude eventual biases between the sensors also in the initial total columns?

*For trend calculations a constant bias (in clouds and ozone) is not really an issue and can be removed using a suitable merging approach as shown here. In the periods of overlaps both total ozone and tropospheric columns agree well after applying a bias correction. In particular the lack of significant drifts in the comparison between GOME-2A and SCIAMACHY over an extended period show that the data records are quite stable. A time-varying bias (drift), however, may add significantly to trend uncertainties if not properly accounted for.*

*In the text we mention: "Possible reasons for the biases are the different cloud algorithms used for each instrument (SACURA for SCIAMACHY and FRESCO for GOME and GOME-2) and the small biases noticed in the total ozone columns (e.g. ∼ -2.5 DU between SCIAMACHY and GOME-2). Differences in spatial resolution and overpass time of the instruments have also minor contributions in the biases."*

Is there any explanation for the different behavior of GOME-SCIA differences over $10^{o}$N shown in Figure 1?

*The larger variation in the bias with latitude in GOME data is most likely due to the short overlap period (10 months, from August 2002 to June 2003, when GOME lost its global coverage). For GOME-2 the overlap with SCIAMACHY was more than 5 years, making the latitude dependence smoother. It should be noted that the shift in the bias at 10°N is within the uncertainty of the observed biases at these latitudes.*

*In the manuscript (page 5, line 22) it is now mentioned that: "GOME mean biases have stronger latitudinal variability than those of GOME-2. This behavior may be explained by the short time of common operation (Jan. 2002–Jun. 2003) between GOME and SCIAMACHY instruments."*

The authors should also provide an explanation for the GOME-2/SCIA drift. Does this originate from a potential drift in the total columns?

*There seems to be a positive drift in the GOME-2-SCIAMACHY difference (Fig. 1) which is quite small and statistically not significant. One possible explanation are changes in the instrument response function with time (e.g. De Smedt et al, 2012).*

*Reference: De Smedt, I., Van Roozendael, M., Stavrakou, T., Müller, J.-F., Lerot, C., Theys, N., Valks, P., Hao, N., and van der A, R.: Improved retrieval of global tropospheric formaldehyde columns from GOME-2/MetOp-A addressing noise reduction*

*and instrumental degradation issues, Atmos. Meas. Tech., 5, 2933–2949, doi:10.5194/amt-5-2933-2012, 2012.*

Section 2.3.
The discussion of six scenarios in the paper is confusing, since they don't differ substantially concerning the outcome. I think the authors should just describe
here the chosen approach of harmonization.

*We think that this section is one of the most important results of this paper. In the six scenarios we checked different reasonable assumptions on how to handle the differences between the individual instruments. We show here that the hamonisation procedure (merging) is one of the largest error sources of the trends since the trends derived from the various merged dataset show larger differences than the statistical uncertainty of the trend derived from any of the single dataset (one of the six). This is usually neglected in other studies. We, however, shortened that section a bit and moved some of the figures to the supplementary material. The added uncertainty from the merging approach is also discussed in more detail in the summary section.*

Section 3.3.1.
The authors present regional trends in this sections. The choice of the regions to my understanding is based only on the significance of the trends and in a sense this looks like a random choice. Do these regions have some special characteristics that have to do either with prevailing dynamic features or emission sources? In general the discussion here should be improved.

*Indeed we studied regional trends focusing on the regions where the trends are statistically significant across many grid points. As we are not trying to speculate too much on the possible causes (would require substantial modelling efforts) we leave it as is. In the introduction and summary we discuss some of the possible causes of trends and make it clear that long-range transport of tropospheric ozone can contribute to trends in rather remote areas.*

Section 3.3.2.
The authors attribute the positive trends to South Africa and South America to biomass burning. Is there any indication from another source that there is increased biomass burning over the years that can cause such a trend?

*The following text has been added (page 17, line 15): "The burned area in southern tropical Africa increased by 1.8 %/yr during the period 2000 to 2011 (Giglio et al., 2013). Ziemke et al. (2009b) and Wai et al. (2014) estimated that biomass burning can contribute to an increase in tropospheric ozone column by $\sim$20%. Hence, it is very likely that biomass burning could be the origin of the observed ozone increase."*

Section 3.3.3.

The authors show trends over mega-cities in the tropics. The authors should provide a comment why they think a grid-point of 2x5degrees can represent the variability of tropospheric ozone caused by a mega city.

*This Section (3.3.3) has been removed from the manuscript.*

The discussion against NO2 trends as shown in the paper is also not conclusive. Are there studies (modelling or in-situ ones) to support their findings?

*The discussion about NO$_2$ trends has been removed (megacities).*

The authors also compare their results with Heue et al and although the approach is pretty similar there are differences. They should elaborate more here to explain this.

*The comparison with Heue et al. ( 2016) trends for 10 mega cities has been removed from the manuscript.*

---

## Referee Report (RR1)

Review for Atmos. Chem. Phys.

"Harmonisation and trends of 20-years tropical tropospheric ozone data"

Authors: Leventidou, Weber, Eichmann, Burrows, Heue, Thompson, and Johnson

Review submitted: 18 March 2018

General Comments:

This review is of the revised paper. While the authors attempted to address many of the concerns raised by the other reviewer and me, I'm afraid I still have some important concerns about this paper. It is not at all clear that Scenario 1 is the best approach (or better than any of the other 5 scenarios the authors test). As a result, the tropical trend calculations are more uncertain that the paper communicates. Perhaps a Monte Carlo approach is necessary to better characterize the uncertainties in the trend analysis? I'm afraid this paper still needs another round of revisions and should not yet be published in ACP.

Comments on Response to Reviewers:

"…and Hilo is strongly affected by volcanic outgassing resulting in negligible ozone concentrations in the boundary layer." That's not quite true. More accurate would be that the tropospheric ozone data in Hilo often shows interference from $SO_2$. It's not that the ozone isn't there; it's that the ECC measurement doesn't work well in regions with significant $SO_2$ (see Komhyr, 1969 and Morris et al., 2010).

"Tropical troposphere ozone trends critically depend on the merging/harmonization approach." Agreed. But a problem for the authors is that they have not shown any one approach is better than another!

Lines 13-14: "…was applied for GOME, and mean biases…were calculated and applied…"

Lines 21 – 21: Since you're citing "decreases," I believe you do not need negative signs in front of the magnitudes.

Page 1 – second one

Line 10: "...during the monsoon period…"

Line 15: "…on the order of…"

Line 16: You repeat "year-1."

Line 34ff: This is the first point at which we find out what THIS paper is going to do. It's a lot of introductory text. Might be good to get to this earlier and better integrate how previous studies shape/motivate the need for this one.

Page 2 – second one

Line 13: "…integrated (up to 200 hPa)…"

Line 14 – 15: Rework. Something is not right…

Line 16: Since you've got two sources listed, delete the word "One," and change to "Large sources of uncertainty are…"

Line 28: "…because SCIAMACHY is the only instrument…"

Line 5: Seems like the sentence should end, "…GOME and GOME-2)." Just delete the rest of the sentence.

Line 21: I thought in the response to the reviewers that the authors were going to limit their analysis to 15S – 15N, yet here they reference results in the 17.5 – 20N latitude band. See also Figure 1 on Page 4.. Why not limit every part of this analysis to the more restrictive latitude band?

Line 4: "…with respect to SCIAMACHY is added…"

Line 16: As I remarked in my comment on the response to the reviewers section, this statement on "high $SO_2$ emissions, resulting in negligible ozone concentrations…" is not accurate. See my earlier explanation.

Lines 3 and 4: Is it "ozone sondes" or "ozonesondes." I'd pick the latter.

Line 14: "For these reasons, scenario 1 has been selected…" I can't figure out a good justification here – lots of issues. The next page contains a table of the ways I've sliced and diced the data you provided and upon which you based your decision. Your analysis looked at the mean bias and took the one closest to 0, which led to your choice of Scenario 1. I've added the standard deviation calculation to the calculation of the mean. As you can see, in every scenario, the standard deviations of the data are greater than the mean biases. Thus, I would argue that the differences in the means are statistically insignificant. This approach is not a good one for selection of the best scenario. Furthermore, if we just eliminate American Samoa, the conclusion is not robust: now Scenario 6 is the clear winner by the smallest mean (although again, the standard deviations exceed the mean biases in every case). Better, I think, is looking at

the root-mean-square bias, because you're really interested in which approach produces the smallest magnitude bias on average rather than the smallest mean of the biases (e.g., biases of -10 and + 10 would have a mean bias of 0 but a rms bias of 10). If you use this approach (which I believe marginally better), you conclude that Scenario 2 has the lowest mean bias, but again not statistically significantly different from any of the other scenario means. Finally, if you eliminate American Samoa, Scenarios 1, 2, 5 and 6 are all pretty close.

| | Scenario 1 | Scenario 2 | Scenario 3 | Scenario 4 | Scenario 5 | Scenario 6 |
|---|---|---|---|---|---|---|
| Am Samoa | -0.89 | -0.92 | -1.99 | -0.61 | -0.93 | 4.59 |
| Ascension | 0.03 | -0.14 | -0.77 | -0.42 | -0.6 | 0.03 |
| Java | -0.11 | -0.12 | -1.12 | -0.54 | -0.55 | -0.11 |
| Kual Lupmur | -1.81 | -2.12 | -2.12 | -2.14 | -2.48 | -1.78 |
| Nairobi | 1.81 | 1.1 | 1.8 | 1.48 | 0.74 | 1.84 |
| Natal | 0.56 | 0.63 | -0.21 | 0.22 | 0.28 | 0.57 |
| Paramaribo | -2.98 | -2.95 | -3.02 | -4.11 | -4.34 | -0.11 |
| Mean bias | -0.48 | -0.65 | -1.06 | -0.87 | -1.13 | 0.72 |
| Std. Dev. | 1.58 | 1.46 | 1.57 | 1.79 | 1.74 | 2.01 |
| | | | | | | |
| Eliminate Samoa | | | | | | |
| Mean bias | -0.42 | -0.60 | -0.91 | -0.92 | -1.16 | 0.07 |
| Std. dev | 1.72 | 1.59 | 1.66 | 1.96 | 1.91 | 1.17 |
| | | | | | | |
| | | | | | | |
| Root-mean-square | | | | | | |
| Mean bias | 0.87 | 0.84 | 1.34 | 0.90 | 0.93 | 1.49 |
| Std. Dev. | 0.79 | 0.74 | 0.76 | 0.75 | 0.79 | 1.71 |
| | | | | | | |
| Eliminate Samoa | | | | | | |
| Mean bias | 0.86 | 0.82 | 1.20 | 0.96 | 0.93 | 0.87 |
| Std. Dev. | 0.89 | 0.83 | 0.77 | 0.82 | 0.88 | 0.89 |

Line 2 – 3: "However, the biases of each scenario with ozone sondes are very close to each other for every station." I don't see the data in Table 1 supporting this statement. There's great variability in both rows and columns.

Line 8: "…the scenarios that can be confidently rejected are…" I see nothing in Table 1 upon which to base any rejection of one scenario over another.

Line 13: "…has the smallest mean bias with the ozone sondes (-0.4 DU)." As you can see above, this mean bias is not statistically significantly different from any of the other scenarios.

Figure 3: Why does panel f show more area as statistically significant in %/year trends than panel e? I think it's because the criteria in panel e is stricter (exceed the range of all harmonization scenarios), but it's confusing to have these next to one another. Why not use the same criteria? I think f makes a more interesting map than e. But based on my analysis above, I don't see one scenario as preferable to another.

Table 2: I take it these 2 sigma uncertainties are determined by the trend analysis itself and to not include the additional uncertainty resulting from the harmonization choice itself? If that's right, these results look better (and more significant) than they are. Perhaps a better approach would be to use a Monte Carlo analysis that mixes between the scenarios and reflects the uncertainty in the bias of the scenario to figure out the total uncertainty in the trend. At this point, I have little confidence in the quoted uncertainty in this Table and as a result, the associated discussion.

---

## Author Response (AR2)

**Review for Atmos. Chem. Phys.**
**"Harmonisation and trends of 20-years tropical tropospheric ozone data"**
**Authors: Leventidou, Weber, Eichmann, Burrows, Heue, Thompson, and Johnson**
**Review submitted: 18 March 2018**

General Comments:
This review is of the revised paper. While the authors attempted to address many of the concerns raised by the other reviewer and me, I'm afraid I still have some important concerns about this paper. It is not at all clear that Scenario 1 is the best approach (or better than any of the other 5 scenarios the authors test). As a result, the tropical trend calculations are more uncertain that the paper communicates. Perhaps a Monte Carlo approach is necessary to better characterize the uncertainties in the trend analysis? I'm afraid this paper still needs another round of revisions and should not yet be published in ACP.

*Our response:*

*We clearly indicated that tropospheric ozone trends in the tropics using multi-instrument data may vary significantly with the merging approach selected. The differences in the calculated trends (up to 6 DU/decade, see: Fig. 2 (top) in the supplement) can be even larger than the statistical uncertainty (less than 4 DU/decade) of the trend derived from any single merged dataset. This means that the trend uncertainties are in reality larger than given by regression statistics alone. This is the first time that different merging approaches are considered and their possible impact on tropospheric ozone trends investigated. This may be one of the reasons why tropospheric ozone trends from different studies do not agree with each other (see e.g. TOAR report). Monte Carlo simulations might be a good idea to better characterize these uncertainties but we believe that such a study is out of the scope of our paper and it will not change our final result that the mean trend in the tropics (15° S-15° N) is statistically non significant and close to zero.*

Comments on Response to Reviewers:
"…and Hilo is strongly affected by volcanic outgassing resulting in negligible ozone concentrations in the boundary layer." That's not quite true. More accurate would be that the tropospheric ozone data in Hilo often shows interference from SO2. It's not that the ozone isn't there; it's that the ECC measurement doesn't work well in regions with significant SO2 (see Komhyr, 1969 and Morris et al., 2010).

*The phrase: "..and Hilo is strongly affected by volcanic outgassing resulting in negligible ozone concentrations in the boundary layer" has been changed as follows: "..and Hilo is strongly affected by volcanic outgassing which interferes with the ozonesondes' electrochemical concentration cells (ECC), resulting in negligible ozone concentrations being measured in the boundary layer (Morris et al., 2010) ."*

"Tropical troposphere ozone trends critically depend on the merging/harmonization approach." Agreed. But a problem for the authors is that they have not shown any one approach is better than another!

*The selection of the preffered merging scenario has been investigated by calculating the mean biases between six scenarios in seven ozonesonde stations from the SHADOZ network. The comparison, as the Reviewer also mentions, showed that there is no strong*

*indications on selecting one of them according to statistics, instead logical arguments have been used in order to decide. These are the fact that the overlap period between SCIAMACHY and GOME is very short (10 months) in order to use correction offsets used in scenarios 3, 4 and 5. For this reason these scenarios can be rejected. Scenario 6 can also be rejected due to the fact that the drift in GOME-2 correction offset is mostly (81%) statistically not significant. The lack of significant drifts between GOME-2 and SCIAMACHY biases over the overlapping period (5 years) shows that the data records are quite stable. For these reasons and due to the fact that scenario 1 has the smallest mean bias with the ozonesondes (-0.4 DU), lead us to select scenario 1 for merging the datasets.*

Lines 13-14: "…was applied for GOME, and mean biases…were calculated and applied…"
*Changed*

Lines 21 – 21: Since you're citing "decreases," I believe you do not need negative signs in front of the magnitudes.
*Changed*

Page 1 – second one

Line 10: "...during the monsoon period…"
*Changed*

Line 15: "…on the order of…"
*Changed*

Line 16: You repeat "year-1."
*Changed*

Line 34ff: This is the first point at which we find out what THIS paper is going to do. It's a lot of introductory text. Might be good to get to this earlier and better integrate how previous studies shape/motivate the need for this one.

*The main findings and the goal of this paper are already presented in the abstract. The introduction presents the necessary background for tropical tropospheric ozone and trend studies. The last paragraph of the introduction discusses the main goal of this study and the structure of the paper. One sentence has been added in order to better connect the previous paragraphs with the last one and the last paragraph of the introduction is now as follows:*

*"Using a convective clouds differential (CCD) method, developed at the Institute of Environmental Physics (IUP)/University of Bremen and applied to retrievals of total ozone and cloud data from GOME/ERS-2 (1995-2003), SCIAMACHY/Envisat (2002-2012), and GOME-2/MetOp-A (2007-2015), new datasets of monthly mean tropical tropospheric columns of ozone (TTCO) have been created (Leventidou et al., 2016). The main differences between our CCD algorithm and the one developed by Heue et al. (2016) mainly originate from the different corrections that we have applied in the above cloud column calculation of GOME and GOME-2 data and handling of the outlier data (Leventidou et al., 2016). The main goal of this study is to derive long-term trends from our merged CCD tropical tropospheric ozone datasets. In a first*

*step the three satellite data are merged into a consistent long-term dataset. Six possible approaches for merging the data are considered and evaluated by comparisons to SHADOZ ozonesondes and by trend evaluations (Section 2). The comparisons to ozonesonde, among other criteria, are used to identify the preferred merging scenario. The trend evaluation of the six merging scenarios will allow us to roughly estimate the contribution of the merging approach to trend uncertainties. In Section 3 the multiple linear regression model is briefly described. Detailed trend results for the tropics 15S -15N as well as for selected regions are presented in Section 4 for the preferred merged dataset. This paper ends with a summary and discussion (Section 5)."*

Page 2 – second one

Line 13: "…integrated (up to 200 hPa)…"
*Changed*

Line 14 – 15: Rework. Something is not right…
*Changed as: " have been found to be within the uncertainties of the mean biases of 6 DU (1 sigma)"...*

Line 16: Since you've got two sources listed, delete the word "One," and change to "Large sources of uncertainty are…"
*Changed*

Line 28: "…because SCIAMACHY is the only instrument…"
*Changed*

Line 5: Seems like the sentence should end, "…GOME and GOME-2)." Just delete the rest of the sentence.

***The sentence is correct as is.***

Line 21: I thought in the response to the reviewers that the authors were going to limit their analysis to 15S – 15N, yet here they reference results in the 17.5 – 20N latitude band. See also Figure 1 on Page 4.. Why not limit every part of this analysis to the more restrictive latitude band?

***Our tropical tropospheric ozone dataset extends to 20° S and 20° N. In order to investigate the biases with which we are going to harmonize the three individual datasets we calculated the mean biases and their drifts (only for SCIAMACHY-GOME-2) using SCIAMACHY as reference throughout these latitude bands. This has been also done by other studies (Heue et al., 2016). However, whenever we refer to results above 15° we mention that their interpretation should be generally made with a cautionary note because they are influenced by the low sampling of data. Later, we show that the under-sampling that we see in the biases appear for some scenarios also in the trends. Only for the mean tropical tropospheric ozone trend we use the data between 15° S and N which are not affected by this effect.***

Line 4: "…with respect to SCIAMACHY is added…"
*Changed*

Line 16: As I remarked in my comment on the response to the reviewers section, this statement on "high SO2 emissions, resulting in negligible ozone concentrations…" is not accurate. See my earlier explanation.
*See our comments above*

Lines 3 and 4: Is it "ozone sondes" or "ozonesondes." I'd pick the latter.
***Changed throughout the text***

Line 14: "For these reasons, scenario 1 has been selected…" I can't figure out a good justification here – lots of issues. The next page contains a table of the ways I've sliced and diced the data you provided and upon which you based your decision. Your analysis looked at the mean bias and took the one closest to 0, which led to your choice of Scenario 1. I've added the standard deviation calculation to the calculation of the mean. As you can see, in every scenario, the standard deviations of the data are greater than the mean biases. Thus, I would argue that the differences in the means are statistically insignificant. This approach is not a good one for selection of the best scenario. Furthermore, if we just eliminate American Samoa, the conclusion is not robust: now Scenario 6 is the clear winner by the smallest mean (although again, the
standard deviations exceed the mean biases in every case). Better, I think, is looking at the root-mean-square bias, because you're really interested in which approach produces the smallest magnitude bias on average rather than the smallest mean of the biases (e.g., biases of -10 and + 10 would have a mean bias of 0 but a rms bias of 10). If you use this approach (which I believe marginally better), you conclude that Scenario 2 has the lowest mean bias, but again not statistically significantly different from any of the other scenario means. Finally, if you eliminate American Samoa, Scenarios 1, 2, 5 and 6 are all pretty close.

*The comparison with the ozone profiles from seven ozonesonde stations and the statistics performed by us does indeed not provide a strong evidence upon which scenario should be selected or rejected. The fact that the ozonesonde profiles represent localized concentrations of ozone whereas the CCD retrievals a much broader region of 2.5°x5° should also be taken into account.*
*In the text we mention: "Although the comparison between the TTCO from the individual harmonised scenarios and the ozonesonde data does not favor clearly any harmonisation scenario, the scenarios that can be confidently rejected are scenarios 3, 4 and 5 where GOME data are corrected with respect to SCIAMACHY since the overlap period between GOME and SCIAMACHY is very short (10 months, 8/2002-6/2003). Scenario 6 can also be rejected due to the fact 10 that the drift in GOME-2 correction offset at 81% of the grid-boxes is statistically non significant. Lack of significant drifts in the comparison between GOME-2 and SCIAMACHY over the overlapping period shows that the data records are quite stable."*
*The last sentence has been changed as follows:*
*"For these reasons, scenario1 (no drift corrections and bias correction for GOME-2) which also has the smallest mean bias with the ozonesondes (-0.4 DU) has been selected to be the preferred harmonisation scenario for merging the TTCO datasets."*
*Therefore, the selection of the preferred scenario is made following logical arguments. The comparison with the ozonesondes has only an auxiliary role in this decision.*

Line 2 – 3: "However, the biases of each scenario with ozone sondes are very close to each other for every station." I don't see the data in Table 1 supporting this statement. There's great variability in both rows and columns.

*The biases between CCD and ozonesondes from the six scenarios are on the order of ±1 DU. Taking into account the differences in the spatial resolution between the CCD retrievals and the ozonesonde profiles and the fact that the uncertainties are on the same order, we consider that these biases are very close to each other and can not help alone in making a decision with confidence.*

Line 8: "…the scenarios that can be confidently rejected are…" I see nothing in Table 1 upon which to base any rejection of one scenario over another.

*Since we have shown that the comparison with ozonesode profiles does not lead to a clear preference for any scenario we make the selection based on other scientific arguments.*

Line 13: "…has the smallest mean bias with the ozone sondes (-0.4 DU)." As you can see above, this mean bias is not statistically significantly different from any of the other scenarios.

*This is true, but the bias with the ozoneseondes is not the only criterion that we used for our decision.*

Figure 3: Why does panel f show more area as statistically significant in %/year trends than panel e? I think it's because the criteria in panel e is stricter (exceed the range of all harmonization scenarios), but it's confusing to have these next to one another. Why not use the same criteria? I think f makes a more interesting map than e. But based on my analysis above, I don't see one scenario as preferable to another.

*Fig. 3 e shows only the pixels where the 2 sigma (statistical) uncertainty of the trend exceeds the maximum absolute difference of the trends calculated between any pairs out of the six scenarios. With this figure we want to highlight how small the number of pixels is where we can state with confidence that the trend is significant.*

Table 2: I take it these 2 sigma uncertainties are determined by the trend analysis itself and to not include the additional uncertainty resulting from the harmonization choice itself? If that's right, these results look better (and more significant) than they are. Perhaps a better approach would be to use a Monte Carlo analysis that mixes between the scenarios and reflects the uncertainty in the bias of the scenario to figure out the total uncertainty in the trend. At this point, I have little confidence in the quoted uncertainty in this Table and as a result, the associated discussion.

*Yes, the 2 sigma refer to the trends calculated from the individual merged datasets. This is "the standard" methodology used also from previous studies in tropospheric ozone trends ( Ziemke et al., 2005; Kulkarni et al., 2010; Heue et al., 2016; Ebojie et al., 2016). As we have mentioned before, we not only estimate the trends and their uncertainties based on one merged scenario, but we estimate the additional uncertainty (may exceed 4 DU/decade) that the merging procedure can introduce to the trends. As the reviewer correctly mentions, the statistical uncertainties of the trends are always lower than if all uncertainties (incl. merging approach) were accounted for. This is exactly the message that our paper wants to communicate.*

*Below follows the track changed manuscript:*

[revised manuscript text omitted]

---

## Author Response (AR3)

**Co-Editor Decision: Publish subject to minor revisions (review by editor)** (11 May 2018) by Yugo Kanaya

Comments to the Author:
Dear Authors,

I find that most of concerns and points raised by the reviewers were adequately addressed. I understand that choosing one best harmonization scenario distinguished from others is difficult because of uncertainties as carefully discussed. Rather, acknowledging such harmonization uncertainty in the analysis is a new and important point by itself. In this context the manuscript should be regarded valuable. I now request minor revisions on the following points:
* * *
*Our response:*

1. page 9 (author response version 2).
Why scenario 1 is superior to scenario 2 should be briefly mentioned.
How are the following results and discussion altered when scenario 2 is selected?

*The following text has been added:*

*Scenarios 1 and 2 have the smallest bias with respect ozonesondes, however both show some differences in regional trends with scenario 1 having larger regions with statistically significant trends, while scenario 2 shows mostly zero trends across the tropics within their uncertainties (see: supplement). The main difference between scenario 2 and one is that the former uses a single (global) bias correction for GOME-2, while scenario 1 biases are corrected individually for each grid-box, which we believe makes physically more sense given the large overlap period. For these reasons, scenario1 (no drift corrections and bias correction for GOME-2) which also has the smallest mean bias with the ozonesondes (-0.4 DU) has been selected to be the preferred harmonisation scenario for merging the TTCO datasets.*

2. page 17, line 7. HO2
*Changed*

3. page 17, line 17. boreal autumn
*Changed*

4. Degree sign is sometimes missing.

**Where "increased/decreased by" or "in the order of" are mentioned in the text, the sign is emitted. In other cases the negative sign is included before the number.**

5. page 18, line 24. Where in main text did the authors discuss the trend -0.1+/-0.3% year-1 mentioned here?
"The global mean trend" should be for global tropics?
*Changed as:*

[revised manuscript text omitted]

---

## Author Response (AR4)

**Answer to Editor #2**
**1. page 17, line 7. Still OH2. HO2 is correct.**

*We have changed the $OH_2$ to $HO_2$.*

**2. Degree sign that I commented is for latitude/longitude. For example page 18, line 22 (15S and 15N). Check throughout your manuscript.**

*We have added a negative sign throughout the document where °S and °W are refereed.*

[revised manuscript text omitted]